# HIV-1 release requires Nef-induced caspase activation

**Jason Segura[1], Joanna Ireland[1], Zhongcheng Zou[1], Gwynne Roth[1], Julianna Buchwald[1], Thomas J. Shen[1], Elizabeth Fischer[2], Susan Moir[3], Tae-Wook Chun[3], Peter D. Sun[1] ***

1 Laboratory of Immunogenetics, National Institute of Allergy and Infectious Diseases, National Institutes of Health, Rockville, Maryland, United States of America, 2 Research Technology Branch, National Institute of Allergy and Infectious Diseases, National Institutes of Health, Hamilton, Montana, United States of America, 3 Laboratory of Immunoregulation, National Institute of Allergy and Infectious Diseases, National Institutes of Health, Bethesda, Maryland, United States of America

* psun@nih.gov

## Abstract

HIV infection remains incurable to date and there are no compounds targeted at the viral release. We show here HIV viral release is not spontaneous, rather requires caspases activation and shedding of its adhesion receptor, CD62L. Blocking the caspases activation caused virion tethering by CD62L and the release of deficient viruses. Not only productive experimental HIV infections require caspases activation for viral release, HIV release from both viremic and aviremic patient-derived CD4 T cells also require caspase activation, suggesting HIV release from cellular viral reservoirs depends on apoptotic shedding of the adhesion receptor. Further transcriptomic analysis of HIV infected CD4 T cells showed a direct contribution of HIV accessory gene Nef to apoptotic caspases activation. Current HIV cure focuses on the elimination of latent cellular HIV reservoirs that are resistant to infection-induced cell death. This has led to therapeutic strategies to stimulate T cell apoptosis in a "kick and kill" approach. Our current work has shifted the paradigm on HIV-induced apoptosis and suggests such approach would risk to induce HIV release and thus be counterproductive. Instead, our study supports targeting of viral reservoir release by inhibiting of caspases activation.

**Data Availability Statement:** RNAseq data are deposited in NCBI Gene Expression Omnibus (accession GSE218175). All other data generated or analyzed during this study are included in this

## Introduction

The use of highly active antiretroviral therapy (ART) is effective in suppressing HIV-1 viremia but insufficient to eradicate the virus from infected individuals. The obstacle in achieving a sterile cure has been largely attributed to the persistence of HIV-1 cellular reservoirs [1]. Hence, much of the current research toward an HIV cure is directed at the removal of these viral reservoirs through 'shock-and-kill' strategies using latency-reversal agents (LRA) [2]. Together, the existing approaches, including the potential use of CAR-T cell therapy [3], has yet to achieve the goal of viral clearance or long lasting suppression of viremia in patients following the removal of ART.

HIV-1 infection activates caspase-mediated apoptotic pathways leading to the death of infected T cells [4–6]. It is widely believed that such apoptotic activation benefits the host as

published article and its Supporting Information files.

**Funding:** All authors were supported in part by the National Institutes of Health Strategic Fund in HIV/ AIDS research from Office of AIDS Research to P. S., and by the Intramural Research Program of National Institute of Allergy and Infectious Diseases, National Institutes of Health to P.S under project number AI-000880.

**Competing interests:** The authors have declared that no competing interests exist.

part of the immune response to eliminate infected CD4+ T cells [7, 8]. Consistently, compounds stimulating apoptotic pathways have been proposed to sensitize latently infected cells for apoptosis [9]. Such interpretation, however, seems evolutionary counterproductive as it offers no benefit to the survival of the virus. Despite an abundance of publications on HIV-induced caspase activation [10], the fundamental question of why HIV induces caspase activation remains a mystery. Caspases are not known to directly benefit HIV-1 viral dissemination, though the transcription of HIV-LTR could be activated by a viral protease cleavage fragment of caspase 8 via NF-κB [11].

Tetherin (BST2) and T cell immunoglobulin and mucin domain proteins (TIM) are known to restrict HIV-1 viral release [12–14]. They are part of host cellular antiviral response to control HIV infection. Much poorly understood, however, are the host genes induced by HIV infection to facilitate the viral release. Recently, we reported that HIV-1 release from infected CD4+ T cells required shedding of adhesion receptor L-selectin/CD62L by members of a disintegrin and metalloproteinases (ADAM) [15]. ADAMs are transmembrane metalloproteinases involved in proteolytic activation of cytokines and cell surface receptors [16]. Their role in tumor metastasis is well established [17], but their involvement in viral infections is new. Consequently, the pathway induced by HIV infection leading to ADAM activation and CD62L shedding for viral release remains undefined.

Here, we show that HIV-1 infected CD4+ T cells undergo preferential activation of caspases resulting in increased PS exchange and loss of CD62L expression on infected cells. The viral-induced loss of CD62L involved multiple caspases as only pan-caspase inhibitors consistently inhibited CD62L expression loss and suppressed the viral infection. Caspase inhibition did not affect the viral replication, but instead, resulted in the accumulation of virions on infected CD4 + T cells. The budding viruses in the presence of the caspase inhibitors consisted of many smaller and aggregated virus-like particles with detectable tethering by CD62L as evidenced from electron microscopy. Consistently, caspase inhibition significantly decreased the structural fitness of released virions. Further, viral release from CD4+ T cells of chronically infected individuals and from their viral reservoirs also requires caspase activation. We further sought to identify the viral mechanism inducing the inflammatory pathway responsible for CD62L shedding. We show that HIV accessory protein, negative regulatory factor (Nef) is important in inducing caspase activation for viral release. Together, we conclude that HIV-1 viral release is not spontaneous but rather requires inflammatory caspase activation. Hence, the primary reason for HIV-1 to induce apoptosis in infected CD4+ T cells is to benefit its own dissemination through release. While the current approach favors inducing apoptosis of infected CD4+ T cells, paradoxically, our results suggest this may lead to increased viral release and thus be counterproductive. Instead, we propose a new antiviral approach based on inhibition of HIV-1 release from both productively infected cells and persistent viral reservoirs by targeting caspase-mediated ADAM metalloproteinase activations. These HIV-1 release inhibitors constitute a new class of anti-viral compound that could synergize with the existing ART regiments toward eliminating persistent viral reservoirs and achieving long lasting suppression of viremia.

## Results

### HIV-1 activates caspases in infected CD4+ T cells

We showed previously that HIV-1 infection induced shedding of its adhesion receptor CD62L/L-selectin, as evidenced by the loss of surface expression of CD62L on infected T cells and the increase of soluble CD62L in infected supernatants [15]. The pathway leading to the viral activation of CD62L shedding, however, remains unknown. As ADAM metalloproteinases are often involved in CD62L shedding [18–20], we first examined if HIV-1 infection

enhanced enzymatic cleavage of CD62L by cell-associated metalloproteinases. To do this, we developed a fluorescence-based enzymatic assay to measure metalloproteinase cleavage of a membrane proximal CD62L peptide substrate. As expected, recombinant ADAM 10 cleaved the CD62L peptide and the peptide cleavage was inhibited by ADAM 10 inhibitors, such as Batimastat (BB-94) and GI254023X [21] (S1A Fig). We then examined the cleavage of the CD62L peptide using cell membrane-associated extracts isolated from a CCR5 tropic HIV-1$_{BAL}$ infected PBMCs as the source of metalloproteinases. The results showed that the membrane extracts from HIV-1 infected samples cleaved CD62L peptide significantly more than those from uninfected samples and the peptide cleavage was inhibited by BB-94 (Fig 1A). Further, soluble CD62L accumulated significantly more in the infected than uninfected supernatants and BB-94 partially inhibited its accumulation (Fig 1A), suggesting the viral infection upregulated metalloproteinase enzymatic activity for CD62L shedding.

HIV-1 infection is known to activate caspases to eliminate infected CD4 T cells but also caused the death of bystander T cells [4]. Interestingly, Caspases are also involved in the activation of ADAM17 [19]. This prompted us to investigate the role of caspases in HIV-1 infection-induced loss of CD62L expression. As expected, analyses of CD4 T cell caspase activation using polyfunctional fluorescent inhibitor of caspases (FLICA) in HIV-1 infected PBMC showed higher caspase activities in HIV-1 infected (p24+) CD4 T cells compared to those in p24- population or uninfected controls (Fig 1B and 1C). The caspase activation coincided with the loss of CD4 and CD62L expressions in the infected but not uninfected CD4 T cells (S1B Fig). To investigate if the viral infection regulated the transcription of caspases, we performed whole cell transcriptome analyses by RNA-Seq of HIV-1$_{BAL}$ infected primary CD4 T cells (Fig 1D, S1 Table). HIV-1 infection upregulated the transcription of antiviral genes, including many interferon-induced and members of TRIM family genes, and downregulated chemokine receptors including CCR5 [22] (S1C, S2A–S2C Figs). In addition, the viral infection upregulated the expressions of many cell signaling genes, such as MAPK, PI3K, PKC, NFAT, and interleukins (Fig 1D, S1C Fig), leading to NF-κB activation and viral gene transcriptions [23]. CD4 T cells express all three classes of caspases and the transcriptions of caspase 2 and 10 (initiator), caspase 1 (inflammatory), and caspase 3 and 7 (executioner) were upregulated during the viral infection (Fig 1E). The transcriptional upregulation of caspases was further confirmed by RT-PCR analyses using primers specific for individual caspases (S2D Fig). The HIV-induced caspase transcriptions are comparable to the upregulation of known HIV restriction factors including SERINC family members, SAMHD, and tetherin (BST2) (S2C Fig) [12, 24–26]. To address if HIV-1 infection resulted in functional activation of individual caspases, we stained the infected CD4 T cells with individual FLICA probes for members of initiator, inflammatory and executioner classes of caspases (S2E Fig). While the majority caspases were activated by the viral infection, caspase 10, the most transcriptionally upregulated isoform, exhibited the most functional activation in infected (p24+) compared to p24- or uninfected populations (Fig 1F). In addition, caspases 1, 8 and 9 were also significantly activated despite no significant transcriptional upregulation were observed. Among the three classes of caspases, members of executioner (caspases 3, 6 and 7) were the least activated by HIV-1 infection (S2F Fig). As the RNA-seq and FLICA analyses were performed using cells from different donors, it is not clear whether the observed lack of caspases 3 and 7 activation and the lack of caspase 8 and 9 transcriptional-upregulation reflect their difference in gene transcription and activation or the donor differences (Fig1E and 1F). Importantly, HIV infection activated all three classes of caspases.

## HIV-1 infection induces phosphatidylserine exposure

Recent work showed that ADAM17 activation on B cells required caspase-induced phosphatidylserine (PS) exposure [19]. While HIV-1 infection resulted in caspases activation and

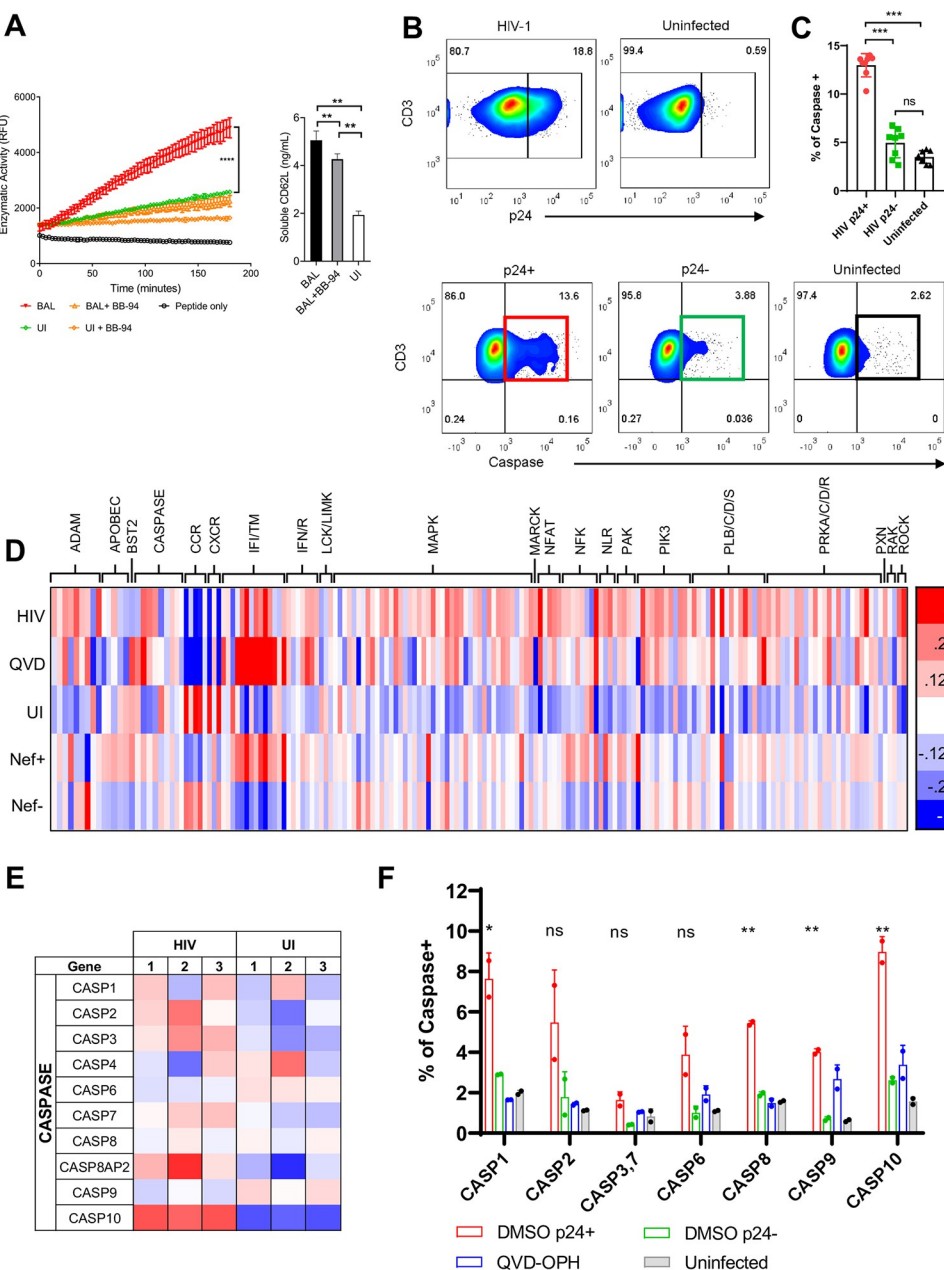

**Fig 1. HIV-1 infection activates caspases and sheddases.** **(A)** (Left panel) Enzymatic cleavage of a membrane proximal CD62L peptide by cell membrane-associated proteases isolated from day 6 HIV-1$_{BAL}$ infected (BAL) or uninfected (UI) CD8-depleted PBMC in the presence and absence of 20 μM BB-94. The data are from technical duplicates and the results are representative of two independent experiments. (Right panel) ELISA measurement of soluble CD62L concentration from the infected (BAL) and uninfected (UI) culture media. The statistics are analyzed using technical repeats in a two-tailed student t-test for each experiment with p values <0.01 (**), p < 0.0001 (****). **(B)** Flow cytometry analysis of caspase activation in CD4+ T cells from HIV-1$_{BAL}$ infected or uninfected PBMC on day 6 of post infection. The cells are stained for intracellular viral capsid protein p24 (top) and activated caspases in p24 + vs p24- T cells (bottom). T cells with activated caspases in p24+ and p24- populations as well as in the uninfected sample are highlighted in red, green and black squares, respectively. **(C)** Bar diagram showing caspase activation in p24 + (red), p24- (green) populations of the infected sample as well as the uninfected control (black). Caspase activation correlated with CD62L and CD4 loss (S1B Fig). Mann-Whitney nonparametric test ***p < 0.001. The results are representative of three independent experiments. **(D)** Differential gene expressions between infected and uninfected CD4+ T cells from representative transcriptome analyses by RNA-Seq. HIV, QVD and UI denote cells infected with HIV-1$_{BAL}$ in the absence or presence of QVD-OPH, or uninfected sample. Nef +/- samples are cells infected with Nef-sufficient or -deficient strains of HIV-1$_{NL4-3}$. HIV-1$_{BAL}$ infection elevated the expressions of many genes involved in

inflammatory signaling pathways. **(E)** Differential gene expressions between HIV-1$_{BAL}$ infected and uninfected CD4 + T cells from 3 independent donors (S1 Table, S2 Fig) showed that the viral infection elevated the expressions of several caspases. **(F)** FLICA staining of individual caspases in primary CD4 T cells from a single donor following infection by HIV-1$_{BAL}$ in the absence or presence of QVD-OPH. Data are presented in duplicates and analyzed by comparing infected DMSO-treated to uninfected controls using student t-test $^*p < 0.05$, $^{**}p < 0.005$.

induced metalloproteinase cleavage of CD62L peptide (Fig 1), it is not known, however, if the viral infection also results in PS exposure in HIV infected T cells. Staining the HIV-1 infected primary T cells with both polyfunctional FLICA and annexin V showed an accumulation of caspases and annexin V double positive population in the infected but not uninfected samples, suggesting the viral infection resulted in caspases activation and PS exposure (Fig 2A). Further, the annexin V positive population in HIV-1 infected samples also showed significant loss of CD62L expression (Fig 2B and 2D), consistent with the notion that the loss of CD62L expression in HIV-1 infected T cells is associated with caspase activation and PS exposure. Interestingly, HIV-1 infected T cells contain both CD62L high and low populations (Fig 2C). Yet, CD62L expression was lost primarily on infected T cells with activated caspases undergoing PS exchange, suggesting the loss of CD62L is the result of caspase activation. Similarly, HIV-1 infected T cells also showed down regulation in CD4 expression (S1B Fig) [15, 27].

In fact, HIV-1 induced PS exposure is a consequence of the infection-induced apoptosis, as similar PS exposure accompanied with the decrease of CD62L expression occurred in camptothecin (CPT), a topoisomerase inhibitor, treated CD4 T cells in the absence of infection (S3A and S3B Fig). CPT is known to induce apoptosis in primary CD4 T cells through activation of PKC and caspase signaling [28]. Notably, annexin V and CD62L consistently stained separate CPT treated cells with little overlap between them (S3C Fig), consistent with the notion that the loss of CD62L expression is associated with apoptotic T cells undergoing PS exchange. The involvement of caspases is evident as treatment with a pan-caspase inhibitor QVD-OPH reduced the number of apoptotic CD62L$^-$ T cells in both HIV infection and CPT treatment (Fig 2A–2D, S3A and S3B Fig). Interestingly, Belnacasan, an inhibitor of inflammatory class caspase 1 and 4 (S3D Fig), failed to reduce either infection or CPT-induced PS exposure (Fig 2A–2D, S3A and S3B Fig), demonstrating a similar apoptotic pathway induced by HIV and CPT and suggesting multiple classes of caspases contribute to the infection and CPT induced PS exposure. To address if the caspase-induced PS exchange affected enzymatic cleavage of CD62L during HIV infection, we carried out the enzymatic cleavage of the fluorogenic CD62L peptide using live infected or uninfected cells, that maintains caspase-induced PS distribution, rather than membrane extracts, which lost the plasma membrane integrity and hence scrambled the PS distribution. The on-cell cleavage of the fluorogenic CD62L peptide by HIV infected PBMC was significantly more than that by uninfected controls (Fig 2F), and cleavage was inhibited by caspase inhibitor QVD-OPH and by ADAM inhibitor BB-94. Thus, the caspase-induced PS exposure enhanced the ADAM enzymatic cleavage of CD62L. These data suggest that the viral-induced caspase activation and PS exchange directly contributed to the cleavage of CD62L on infected cells.

## Targeting pan-caspase activation suppressed HIV-1 infection

While HIV-1 infection-induced down regulation of CD62L expression is linked to caspases activation and PS-flipping, it is not clear which caspases contributed to the down regulation of CD62L expression as most of the caspases were activated by the viral infection (Fig 1F). To address this, we infected CD8-depleted PBMC's with HIV-1$_{BAL}$ in the presence of individual caspase inhibitors, specific to caspase 2, 3, 8 or 9. The results showed that none of the single

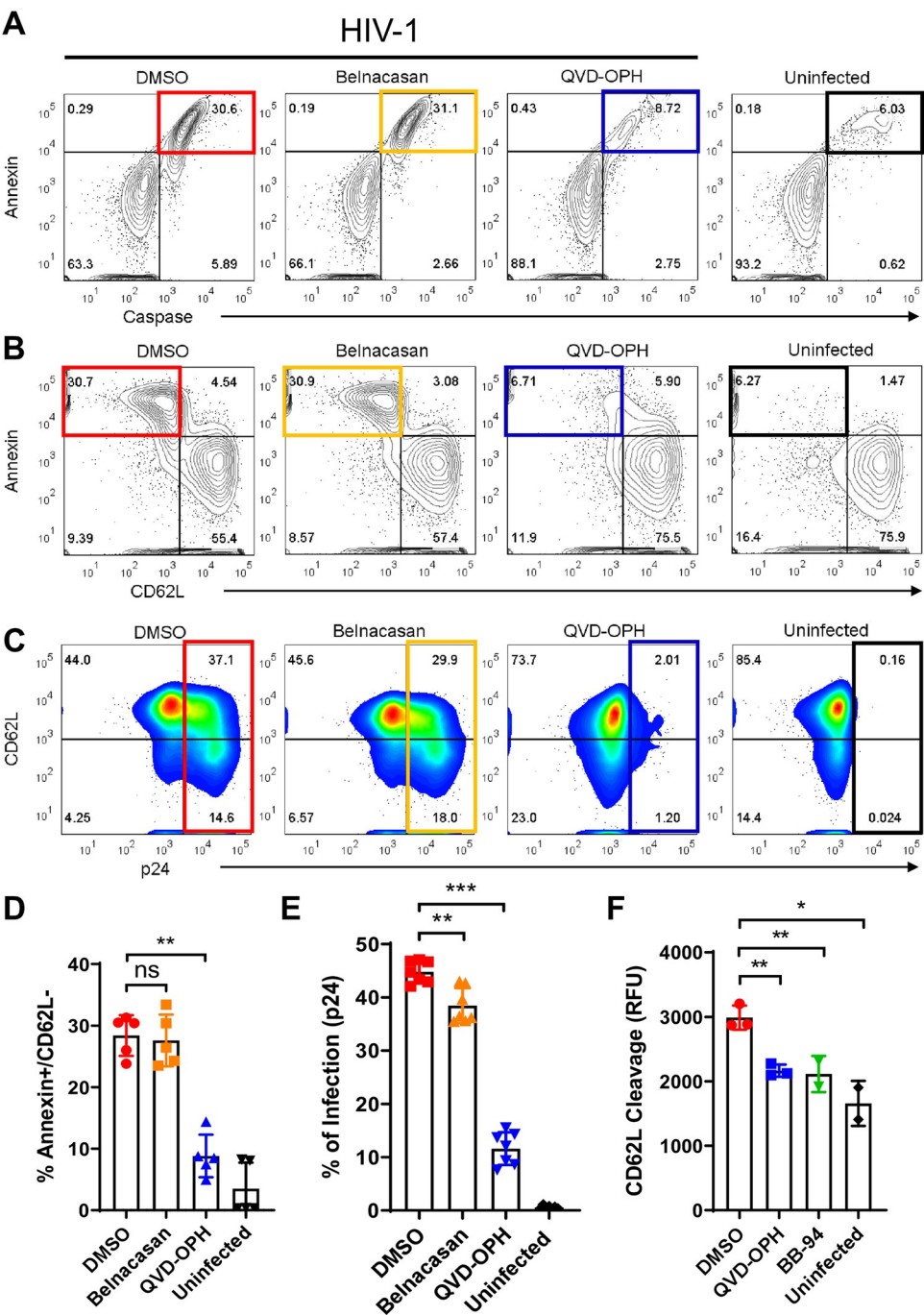

**Fig 2. Caspase activation results in phosphatidylserine exposure and loss of CD62L expression in infected T cells.**
(A-C) Representative FACS contour plots on day 6 HIV-1$_{BAL}$ infected PBMC stained with anti-p24 antibody, annexin V, fluorescent inhibitor of caspases (FLICA), and anti-CD62L antibody. The populations exhibiting caspase+/annexin V+ (**A**) and annexin V+/CD62L- (**B**) are highlighted in colored quadrants. QVD-OPH (blue gates) but not Belnacasan (orange gates) inhibited caspase activation (**A**), CD62L loss (**B**) and the viral infection (**C**). Histograms corresponding to the FACS data presented in panels A and B are shown in S3E Fig for the caspases and CD62L expressions as well as annexin V staining in the infected and uninfected T cells. (D, E) Quantification of the populations in panels **B** and **C**. (**F**) On cell enzymatic cleavage of a fluorogenic CD62L peptide by HIV-1$_{BAL}$ infected and uninfected PBMC in the presence of QVD-OPH, BB-94 or control DMSO. Data are representative of three independent experiments and statistics are shown as Mann-Whitney nonparametric test $^*$p $<$ 0.05, $^{**}$p $<$ 0.01.

caspase inhibitors affected the loss of CD62L in infected (p24+) T cells nor did they inhibited the viral infection (Fig 3A and 3B). Similarly, the presence of oligo-caspase specific inhibitors, targeted at caspases 1 and 4 (Belnacasan), 3 and 7 or 3, 6, 7, and 10, as well as their combination, resulted in at best marginal inhibitions to both the down-regulation of CD62L and the viral infections (Fig 3A and 3B, S4A and S4B Fig). In contrast, the presence of a pan-caspase inhibitor, QVD-OPH [29, 30], significantly reduced the accumulation of CD62L⁻ infected T cells and suppressed the viral infection (Figs 2, 3A and 3B), consistent with the involvement of multiple compensatory caspases in HIV infections. Likewise, the infections of primary T cells by a CXCR4 -tropic virus (HIV-1$_{LAI}$) were also significantly suppressed only by pan-caspase inhibitors ZVAD-FMK and QVD-OPH but not by Belnacasan (Fig 3C), suggesting caspase activation is required for both R5- and X4-tropic HIV infections. This is further supported by a linear correlation between caspase staining and the viral p24 staining from multiple HIV-1$_{BAL}$ infection experiments (Fig 3D). Both QVD-OPH and ZVAD-FMK exhibited a dose-dependent suppression of infection with no significant adverse effect on cell toxicity nor the expressions of CD4 and CD62L (S4C–S4F Fig). This requirement of caspase activation appears to be specific for HIV as QVD-OPH did not suppress Vesicular stomatitis virus (VSV) infections of PBMC (S4G Fig), suggesting that the activation of caspases is not generally required for viral infections, but is specifically required in HIV infections.

HIV-1 infection is known to spread through cell-to-cell transfers [31]. To evaluate the effect of caspase inhibition on cell-to-cell contact mediated viral transmission, we assessed the lateral transmission of virus from infected primary PBMC to target TZM-BL cells in the presence of QVD-OPH following the removal of free virus. As QVD-OPH inhibited the viral infection of PBMC (Fig 3B and 3C), a decrease in cell-to-cell transfer infection is expected in the presence of the pan-caspase inhibitor. Indeed, while caspase 1 and 4 inhibitor Belnacasan marginally reduced the cell-to-cell transfer infection compared to that of DMSO, the pan-caspase inhibitor, QVD-OPH, markedly suppressed the cell-to-cell infections (Fig 3E).

To address if caspase inhibition also affected cell-to-cell viral transfer efficiency, we further calculated the cell-to-cell transfer-infection ratio between successive dilutions of PBMC (4:1 or 2:1) and that of 1:1 (Fig 3F). If a treatment does not affect the efficiency of viral transfer from PBMC to TZM-BL, the transfer-infection ratio should remain similar between treated and untreated samples. Indeed, Belnacasan did not affect the cell-to-cell viral transfer compared to the control despite a partial reduction in cell-to-cell transfer infections were observed throughout the PBMC:TZM-BL titration range. In contrast, QVD-OPH significantly decreased the cell-to-cell transfer efficiency in both 4:1 (PBMC:TZM-BL) and 2:1 cell-to-cell transfers (Fig 3F). Thus, caspase activity is also required for viral transmission between cells.

Mechanistically, QVD-OPH did not inhibit the viral entry of either R5- or X4-tropic pseudoviruses (Fig 4A). Despite a significant reduction in the total copies of viral RNA (Fig 4B), QVD-OPH did not affect HIV transcription as measured by the cellular content of viral RNA normalized against their genomic DNA compared to the controls (Fig 4C). Note, QVD-OPH did not suppress the transcription of ADAM and caspase genes as analyzed by either RNA sequencing or RT-PCR of infected samples albeit partial suppression of ADAM8 and 19 were observed (Fig 4D and 4E), suggesting that the caspase inhibitor affected specifically caspase enzymatic activities but not their expressions.

## Caspase inhibition prevented HIV-1 release and reduced viral fitness

As caspase inhibition did not affect HIV-1 entry and transcription, we then investigated if caspase inhibition suppressed HIV-1 release. On day 6 of post infections treated with Belnacasan, QVD-OPH, ADAM10/17 inhibitor BB-94, or control DMSO, we harvested both viruses

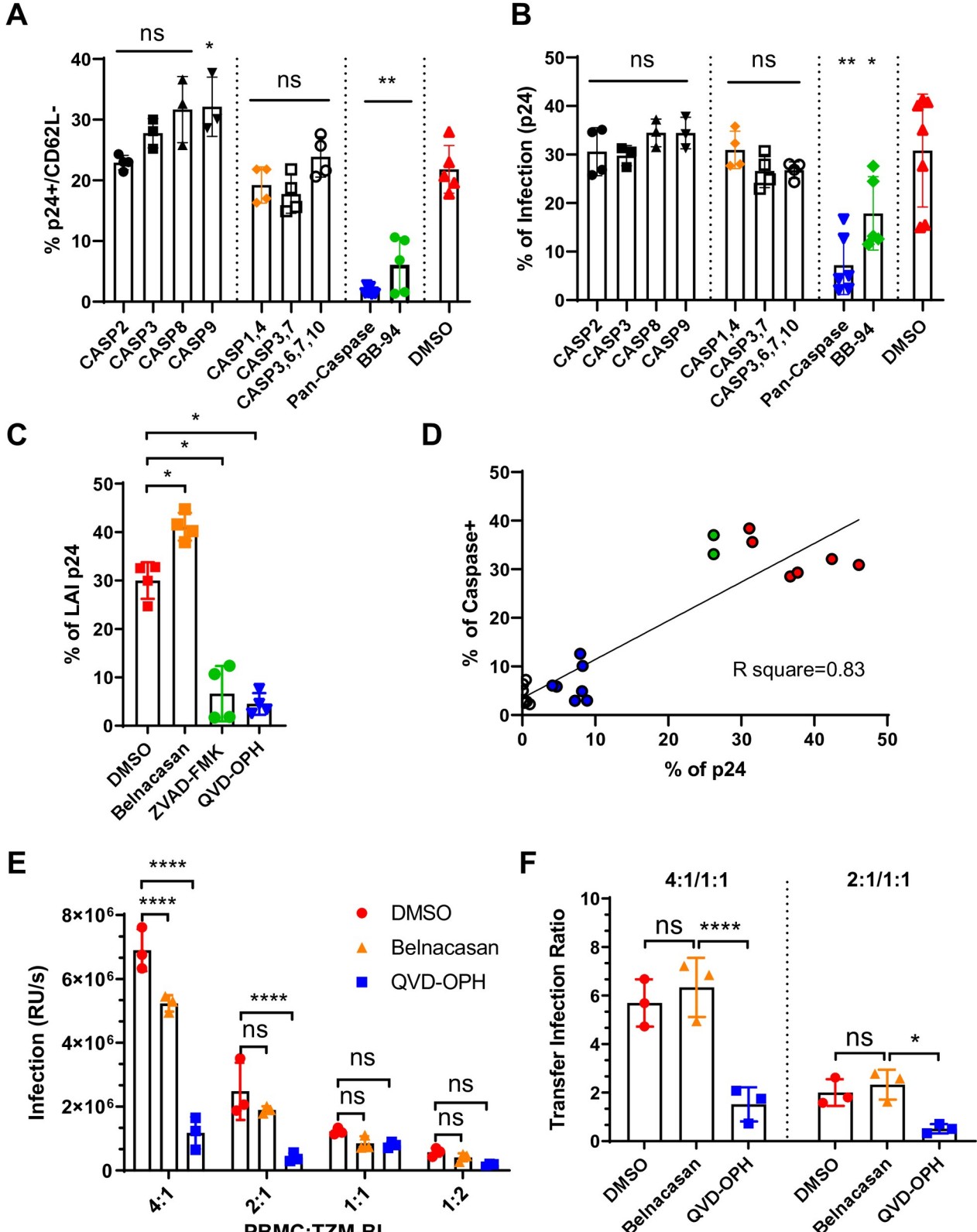

**Fig 3. Targeting pan-caspase activation suppresses HIV-1 infection. (A-B)** FACS analyses for HIV-1$_{BAL}$ infection level and the loss of CD62L expression in infected CD4 T cells in the presence of specific caspase inhibitors. Primary CD8-depleted PBMC were infected with HIV-1$_{BAL}$ for 6 days in the presence of 50uM indicated caspase inhibitors and analyzed for the accumulation of p24+/CD62L- populations (**A**) as well as the level of

infection using intracellular p24 staining (**B**). The cells are stained and gated by anti-p24 and anti-CD62L antibodies as well as by FLICA specific to indicated caspases. Single caspase inhibitors used are Z-VDVAD-FMK (casp 2), Z-DQMD-FMK (casp 3), Z-IETD-FMK (casp 8), and Z-LEHD-FMK (casp 9). Oligo-caspase inhibitors used are Belnacasan (casp 1 and 4), Ac-DEVD-CHO (casp 3 and 7), and Z-DEVD-FMK (casp 3,6,7 and 10). QVD-OPH was used as the pan-caspase inhibitor. The results are from two independent experiments. All statistics were calculated with respect to the control DMSO-treated experiments using Mann-Whitney nonparametric test $^*p < 0.05$, $^{**}p < 0.01$. (**C**) Bar diagram showing suppression of an X4-tropic HIV-1$_{LAI}$ infection of PBMC by pan-caspase inhibitors, ZVAD-FMK and QVD-OPH. The results are representative of two independent experiments analyzed on day 6 of post infection. Mann-Whitney nonparametric test $^*p < 0.05$. (**D**) Correlation between caspase activation and infection level. Those treated with DMSO (red) or Belnacasan (green) had the higher infections and caspase activations than QVD-OPH treated (blue). The uninfected controls are shown in open circles. The linear regression resulted in an R square of 0.83 (goodness of fit). (**E**) Inhibition of cell-to-cell transfer mediated viral transmission. HIV-1$_{BAL}$ infected PBMC were incubated with TZM-BL reporter cells at various density ratios in the presence of Belnacasan, QVD-OPH or DMSO control. The infections are measured luciferase activities from the reporter cells. 2-way ANOVA using Tukey's method performed on triplicates of each compound per culture ratio. $^{****}p < 0.0001$. (**F**) Bar diagram showing the ratio of infections between 4:1, 2:1 (PBMC to TZM-BL), and that of 1:1 transfer infections. The efficiency of viral transfer following removal of free virus is significantly decreased for QVD-OPH but not Belnacasan. 2-way ANOVA using Tukey's method performed on triplicates of each compound $^*p < 0.05$, $^{****}p < 0.0001$.

released in the media and those attached on infected cell surface that were further recovered by trypsinization, and quantified the infectivity of released virions by infecting TZM-BL reporter cells. While QVD-OPH inhibited both free and cell-associated viral productions (Fig 4F and 4G), the ratio of cell-associated virus versus that released in the media, however, was significantly higher in QVD-OPH than DMSO treated infections (Fig 4H), suggesting the pan-caspase inhibitor reduced viral release. Similar but to a lesser extent, BB-94 but not Belnacasan also resulted in accumulation of virus on infected cell surface.

To visualize the caspase inhibition on HIV viral release, we examined the budding of virus from HIV-1$_{BAL}$ infected CD4+ T cells using both transmission (TEM) and scanning electron microscopy (SEM). HIV-1$_{BAL}$ buds from infected T cells as 100–150 nm particles (Fig 5A and 5B) [32, 33]. Most budding viruses in the control treatment appeared as mature virions with classically shaped capsids clearly visible (Fig 5A). Similar sized virions were observed in the presence of Belnacasan (S5A Fig). In the presence of QVD-OPH, however, we observed the emergence of many diminutive virus-like particles forming aggregated networks on infected cells (Fig 5D and 5E). When virions in each TEM image were further grouped into mature, immature and small classes based on their size and the presence of visible capsid core, we found the fraction of mature virions in the presence of the pan-caspase inhibitors, ZVAD-FMK and QVD-OPH were significantly lower than the DMSO control samples (Fig 5G). Conversely, ZVAD-FMK and QVD-OPH treated samples contained significantly higher percentage of smaller sized virus-like particles, suggesting caspase inhibition increased the number of defective viral progenies. While it remains unclear if these smaller virion-like structures are indeed viral particles, their appearance resemble those on HIV-1$_{BAL}$ infected PBMC in the presence of BB-94, an enzymatic inhibitor of ADAM metalloproteinases [15]. The reduction in mature virions in the presence of the caspase inhibition prompted us to address if caspase inhibition affected viral maturation, we first examined the effect of QVD-OPH to the enzymatic activity of HIV-1 protease and showed that the caspase inhibitor did not affect the viral protease enzymatic activity (S4H Fig). Secondly, we performed western blot analysis on HIV-1$_{BAL}$ infected PBMC to detect p55 Gag as well as the cleaved p24 and p17 proteins in the presence of DMSO, QVD-OPH, or Nelfinavir, a known HIV protease inhibitor (Fig 5I). Both QVD-OPH and Nelfinavir significantly inhibited the viral infection (Fig 5J, S4I Fig), and decreased p24 level both in released viruses and cell lysates from the western blot analysis compared to DMSO (Fig 5I). The efficiency of Gag p55 cleavage by HIV protease can be estimated by the ratio of p24/p55 band intensities in their cell lysates. While Nelfinavir significantly decreased the p24/p55 ratio compared to their DMSO controls, QVD-OPH resulted in similar p24/p55 ratio as DMSO(Fig 5K), supporting the notion that Nelfinavir but not

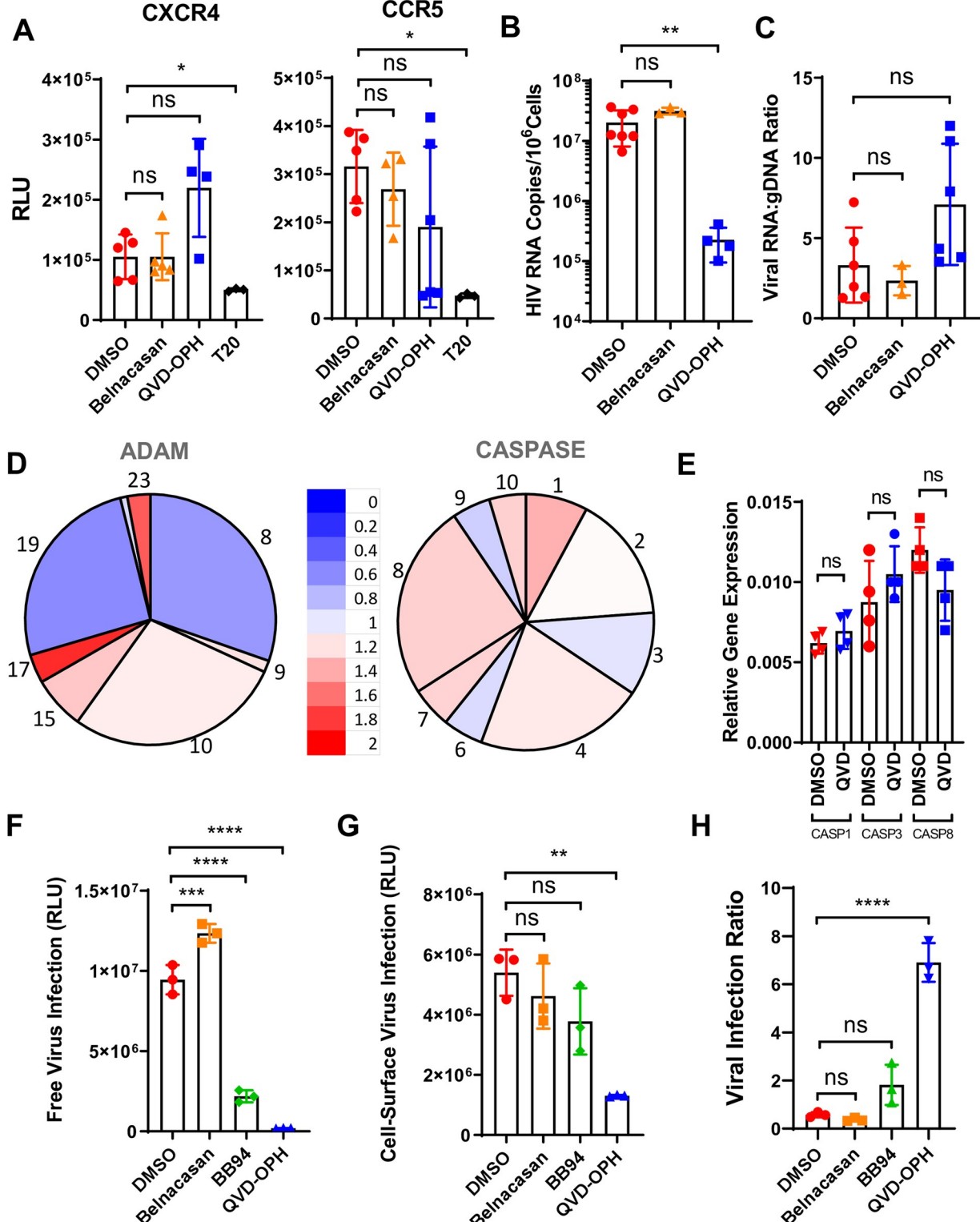

**Fig 4. Caspase inhibition reduced HIV viral release. (A)** Infections of replication-incompetent JRFL (CCR5-tropic) and SF33 (CXCR4-tropic) luciferase-expressing pseudo-HIV to PBMC in the presence of caspases inhibitors, Belnacasan (orange), QVD-OPH (blue), HIV-1 entry inhibitor, T20 (black), or control DMSO (red). The levels of infections are measured by luciferase activities in relative light units. Statistical analyses are performed using Mann-Whitney nonparametric test $^*$p < 0.05. **(B)** Total HIV RNA per $10^6$ cells was determined by RT-PCR of day 6 HIV-1$_{BAL}$ infected primary CD4 T cells in the presence of Belnacasan, QVD-OPH or control DMSO. Significant reduction of total HIV RNA

was observed in QVD-OPH treated infections. **(C)** Viral transcription levels were measured by the ratio between copies of HIV-1 RNA and genomic DNA (gDNA) in day 6 HIV-1$_{BAL}$ infected primary CD4 T cells. HIV gDNA copies were measured with LTR primer by real time PCR using ACH-2 cells that containing a single integrated copy of HIV as standard. The results are from two independent experiments. Mann-Whitney nonparametric test $^{**}$p < 0.01. **(D)** Pie charts showing frequency (size of slice) of gene expressions and their ratio (color of the heatmap) between HIV-1$_{BAL}$ infected primary CD4+ T lymphocytes in the presence and absence of QVD-OPH for ADAM and caspase genes from representative duplicates of RNA sequencing data (Fig 1D). **(E)** RT-PCR analyses of the caspase expressions using caspases 1, 3, and 8 specific primers from day 3 HIV-1$_{BAL}$ infected PBMC in the presence of QVD-OPH or DMSO. The gene expression level is calculated relative to that of control beta actin. **(F-H)** Representative viral release assays from two independent experiments. Total infectious units produced from day 6 HIV-1$_{BAL}$ infected PBMC in the presence of Belnacasan (orange), BB-94 (green), QVD-OPH (blue), or DMSO (red) in the form of both free virus released into media **(F)** and cell surface-associated virus recovered through trypsinization **(G)** were determined using TZM-BL reporter cells. The infectious ratio between cell-associated and free supernatant viruses measures relative viral association on cells **(H).** While BB-94 and QVD-OPH reduced both the number of free released and cell-associated viruses, QVD-OPH treated infections showed proportionally more cell-surface associated viruses. The statistical analyses were performed using one-way ANOVA $^{**}$p < 0.01, $^{***}$p < 0.001, $^{****}$p < 0.0001.

QVD-OPH reduced the viral Gag cleavage. Thus, caspase inhibition is unlikely to affect the viral maturation. As caspase inhibition prevented the loss of CD62L expression (Fig 2B), we then examined whether the presence of CD62L, a C-type lectin receptor known to bind HIV-1 envelope glycan [15], hinders HIV-1 particle release through potential tethering of the viral glycan in the presence of QVD-OPH. To do this, we stained HIV-1$_{BAL}$ infected PBMC with gold-labeled anti-CD62L for TEM analysis and found that the budding virions in the absence of QVD-OPH were primarily segregated from CD62L-expressing cells (Fig 5C). In the presence of QVD-OPH, however, virions became increasingly associated with CD62L (Fig 5F, S5B Fig).

To address if the pan-caspase inhibition resulted in the release of not only fewer, but potentially less infectious viruses, we measured the infectivity of released viruses from infected PBMC using TZM-BL reporter cells over a wide range of viral doses and normalized their infection levels against their viral capsid p24 concentrations measured by ELISA. This capsid-normalized infectivity reflects the infectious quality of progeny viruses. If virions released under caspase inhibition are fewer but normal sized particles without structural deficiency, both p24 concentration and infectivity of the progeny virus would decrease in proportion to the number of virions released with the p24 concentration-normalized infectivity unchanged. However, if the released virions contain structurally deficient non- infectious particles, their capsid-normalized infectivity may decrease. Compared to the DMSO control, the viral progeny from QVD-OPH or BB-94 treated infections exhibited reduced infectivity when normalized against their capsid concentrations in all viral dose range (Fig 5H), suggesting not only fewer but also defected virions were released upon caspase inhibition.

## HIV-1 Nef contributes to caspase activation for viral release

HIV-1 accessory protein Nef is required for efficient viral replication and cytopathic effects in primary cells and for disease progression in humans and animal models [34–39]. Nef contributes to persistent infection of quiescent cells in the presence of ART [40, 41]. HIV-1 Nef is known to activate host cellular signaling pathways, such as PKC pathways resulting in downregulation of MHC-I and CD4 expressions, and caspases-mediated apoptotic pathways [42–49]. More recently, Nef was shown to antagonize TIM-mediated restriction to HIV viral release [14]. Of particular interest here is the role of Nef in viral-induced caspase activation and apoptosis.

To this end, we infected PBMC with HIV-1$_{NL4-3}$ virus containing either wild-type Nef (Nef$^+$) or a stop codon insertion within the Nef ORF paired with initiation codon mutation (Nef$^-$) to address the effect of Nef in caspases activation [50]. Transcriptome analysis by RNA-seq of CD4+ T cells infected in the presence or absence of Nef showed many gene expression differences present between cells infected with the two viruses (Fig 1D). For example, Nef appears to be responsible for the upregulation of interferon inducible genes as well as for the

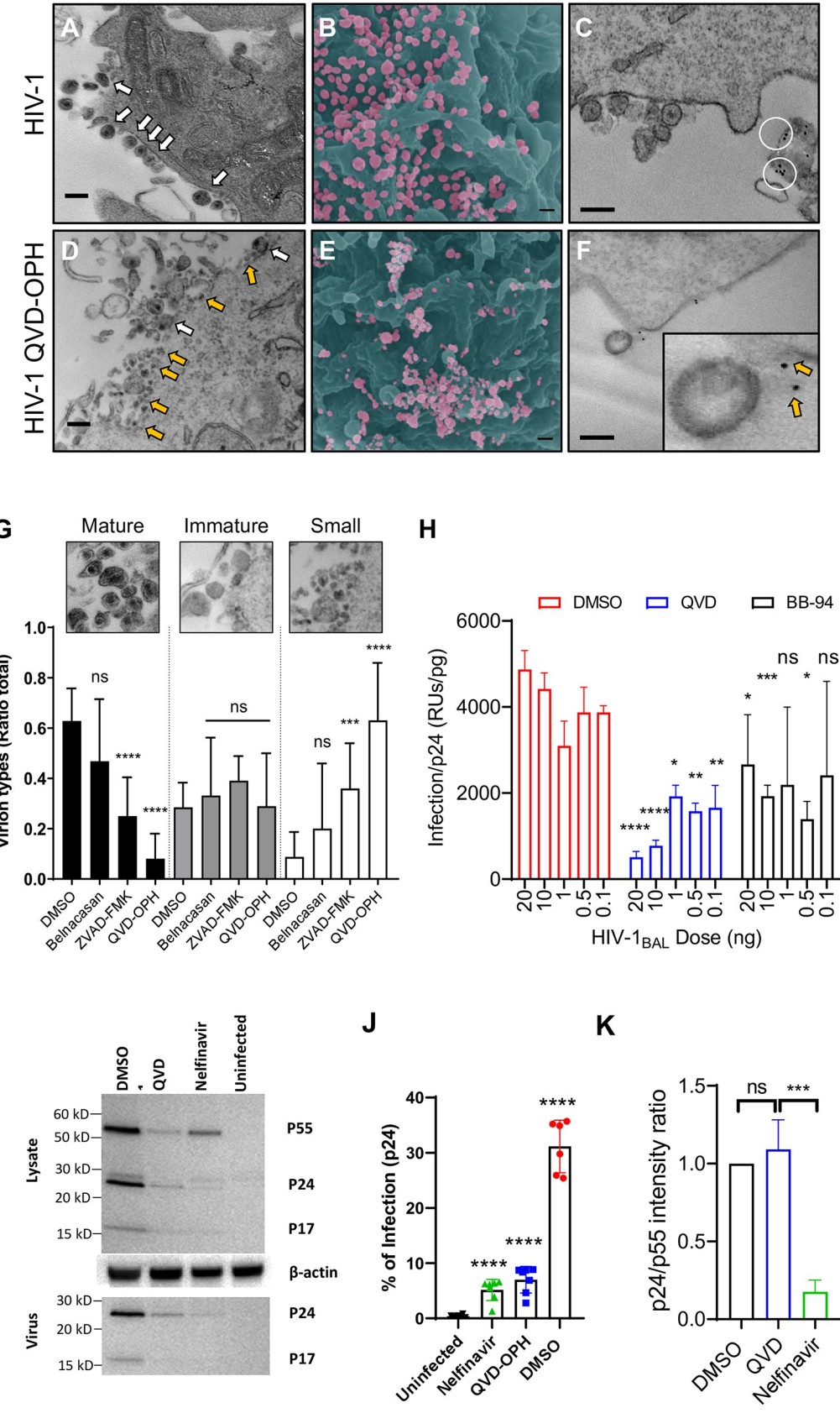

**Fig 5. Caspase inhibition prevents HIV-1 release and reduces viral fitness. (A-F)** Representative electron microscopy images of day 6 HIV-1$_{BAL}$ infected PBMC in the absence (A-C) or presence (D-F) of QVD-OPH. Transmission-EM **(A, D)** and pseudo-colored Scanning-EM **(B, E)** images of budding virions on infected cells. Majority virions in the control treated samples (panels A-C) show a uniform distribution of typical morphological sized particles of ~100–150 nm (white arrows) with visible capsids. Smaller sized (<50 nm) aggregating virus-like particles were frequently observed in QVD-OPH treated samples in both TEM (orange arrows) **(D)** and SEM **(E)** images. Panels C and F are TEM images of day 6 HIV-1$_{BAL}$ infected PBMC stained with anti-CD62L together with a 10nm gold particle-conjugated secondary antibody. Majority virions were found segregated from gold-labeled CD62L in control DMSO treated cells **(C)**, but colocalized in the presence of QVD-OPH **(F)**. All scale bars = 200 nm with the insert in (F) presented in 4x additional magnification. **(G)** Bar diagrams with matching representative EM images show size distribution (mature ≥ 100 nm, immature ≥ 100 nm without capsid, and small ≤ 50 nm) of virus-like particles (VLPs) by ratio of total observed particles associated with primary CD4+ T lymphocytes infected with HIV-1$_{BAL}$ in the presence or absence of QVD-OPH quantified from ~5 images per condition over 3 independent experiments. ***p < 0.001, ****p < 0.0001. Statistical analysis performed on pooled data from all donors with no data omitted. **(H)** Viral fitness assay comparing viral p24 concentration normalized infectious activities among DMSO, QVD-OPH and BB-94 treated infections. The infectious activities of released viral progeny were measured through infection of TZM-BL reporter cells and the concentrations of p24 viral capsid protein in released viruses were quantified by ELISA. Data were presented as triplicates from each treatment and the statistical analysis was done using 2-way ANOVA test *p < 0.05, **p < 0.01, ****p < 0.0001. The results are representative of two independent experiments. **(I)** Immunoblot of gag p55, p24, and p17 proteins in the infected cell lysate (Lysate) or in released free viruses (Virus) from infections treated with DMSO, QVD-OPH, or Nelfinavir. **(J)** FACs analysis of the infection levels on samples used for the immunoblots Compared to DMSO controls, QVD-OPH and Nelfinavir significantly reduced p24 levels in both immunoblots and FACS analyses (panel I and J). The results were representative of four independent experiments. The statistical analysis was performed using one-way ANOVA ***p < 0.001. **(K)** Normalized p24/p55 band intensity ratio to DMSO controls from three independent western blot experiments. Individual p24 and p55 band intensities were determined from cell lysate of the same sample using ImageJ (https://imagej.nih.gov/ij/).

down regulation of chemokine receptor genes (Fig 1D). As caspases are activated by inflammasomes, by intrinsic interferon-induced and extrinsic death receptor-induced apoptotic pathways [51–54], we further analyzed differential transcriptions of genes involved in these pathways between the Nef+ and Nef- infected samples. Indeed, compared to the Nef-deficient virus, Nef+ HIV significantly upregulated the transcriptions of multiple caspase-involved apoptotic pathways, including inflammasome-, and type I interferon-mediated, TNF-α, and p53-dependent apoptotic pathways (Fig 6A, S5G Fig) [55, 56], suggesting Nef directly contributed the activation of multiple caspases. This is also consistent with the observation that only pan-caspase inhibition effectively suppressed HIV infection. In contrast, Nef did not affect the overall transcription of non-apoptotic pathways, such as IL-2 and TLR signaling pathways (S5F Fig).

Consistently, Nef+ HIV infections resulted in significantly higher FLICA staining of caspases than that of Nef- HIV infections with higher caspases staining observed in p24+ than p24- populations in each case (Figs 1C, 6B and 6C). Further, Nef+ HIV infections exhibited increased PS exchange, and a greater number of CD4-/CD62L- T cells than Nef- infections (S5C and S5D Fig). Thus, Nef enhances HIV-induced apoptotic caspases activation. The link between Nef and caspases is further supported by the observation that QVD-OPH reduced the level of caspases staining associated with Nef+ infection to that of Nef- infection (Fig 6D and 6E). Interestingly, Nef- HIV exhibited lower but significant caspases staining and infection of PBMC, suggesting other viral genes may contribute to caspases activation. To address if Nef also contributed to viral replication in addition to caspase activation, we examined the Nef + and Nef- HIV infection to HIV susceptible CD4+ CEM T cells [57]. CD62L is constitutively shed on CEM cells [58], thus avoiding the need for caspase activation. Interestingly, Nef+ and Nef- NL4-3 viruses infected the CEM cells at similar levels (S5E Fig), suggesting Nef does not affect viral replication in the CEM cells. This is consistent with the notion that Nef is often dispensable in HIV propagation in cell lines, and the requirement of Nef in the viral propagation in primary cells is likely due to Nef-induced caspases activation viral release. While the HIV

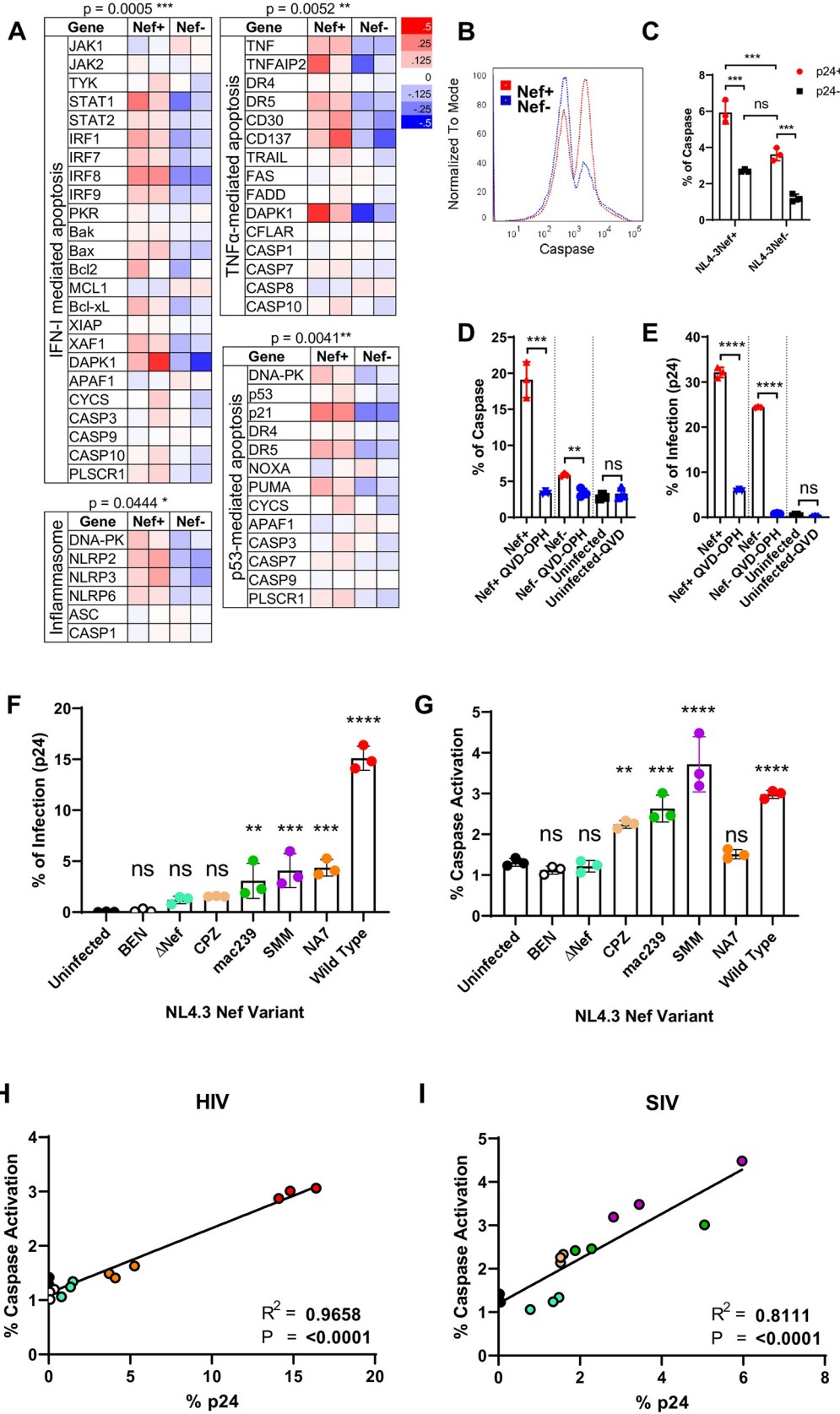

**Fig 6. HIV-1 Nef mediates caspase activation for viral release. (A)** RNAseq analyses for differential gene expression on highlighted gene groups responsible for the cellular signaling pathways leading to caspase activation and apoptosis. RNA sequencing were performed using day 6 Nef+ and Nef- HIV-1$_{NL4-3}$ infected CD4 T cells from 2 donors. Depicted are genes involved in interferon-induced intrinsic apoptotic pathway, TNF-alpha and death receptor-mediated extrinsic apoptotic pathways, as well as genes involved in inflammasome activation. Nef+ HIV infection resulted in significant upregulation of transcriptions of these pathways compared to Nef- infections. Paired student t-test, $^*p < 0.05, ^{**}p < 0.01, ^{***}p < 0.001$. **(B)** Histograms showing total caspase expressions in Nef+ (red) and Nef- (blue) HIV-1$_{NL4-3}$ infected PBMC on day 6 post infections. **(C)** FLICA staining of caspases in p24+ (red) and p24- (black) populations of day 7 infected PBMC using Nef+ and Nef- HIV-1$_{NL4-3}$ viruses. The results are representative of two independent experiments. 2-way ANOVA $^{***}p < 0.001$. **(D-E)** Percentage of total caspase activation **(D)** and p24 staining **(E)** on day 7 of Nef+ or Nef- HIV-1$_{NL4-3}$ infected PBMC in the presence or absence of QVD-OPH. The statistics are calculated using student t-test $^{**}p < 0.01, ^{***}p < 0.001, ^{****}p < 0.0001$. **(F-G)** Level of infection by p24 staining (F) and resulting pan-caspase activation (G) in PBMC's infected by HIV-1$_{NL4-3}$ clones expressing different sequences for Nef. One-way ANOVA $^{**}p < 0.01, ^{***}p < 0.001, ^{****}p < 0.0001$. **(H-I)** Correlation plots between p24 and caspase activation across Nef variants from HIV (H) or SIV (I). $R^2$ and Pearson correlation denoted $^{****}p < 0.0001$.

infection of CEM T cells does not depend on Nef in the absence of caspase inhibition, the viral infection, however, became Nef-dependent in the presence of caspase inhibition (S5E Fig), consistent with Nef's role to counter caspase inhibition.

To further characterize the role of HIV Nef in caspase activation and viral expansion, we infected PBMC with a replication-competent NL4-3 provirus carrying different Nef variant sequences derived from HIV-1 NL4-3 (wildtype), an asymptomatic individual (NA7), HIV-2 (BEN), from chimpanzee (cpz), rhesus macaque (mac239), and sooty mangabey (smm) SIV strains, as well as with the Nef-deletion strain of NL4-3 [50]. Consistently, samples infected with the wildtype Nef-expressing virus exhibited much higher p24 levels as well as higher caspase activations than virus with Nef deleted (ΔNef) (Fig 6F and 6G). Nef from HIV-2 (BEN) and from an asymptomatic individual (NA7) showed significantly lower levels of infections than the wildtype Nef-expressing NL4-3. Both also exhibited little caspase activation comparable to the wildtype, resulting a correlation between caspase activation and their infections (Fig 6H). Interestingly, Nef from all three non-human primates (cpz, mac239, and smm) induced caspases (Fig 6G). This is consistent with previous findings of high viremia observed in non-human primates despite the lack of progression to AIDS [50, 59]. Indeed, both human and simian Nef variants showed good correlation between activation of caspases and their infections (Figs 3, 6H and 6I). This Nef-dependent caspases activation and down regulation of CD62L expression support its involvement in HIV release.

## Caspase inhibition suppressed viral release from HIV reservoirs of infected individuals

The persistence of cellular HIV-1 reservoirs remains one of the greatest obstacles in the ongoing challenge to proclaim a sterilizing cure [60]. The maintenance of HIV reservoirs appears to depend on a dynamic relationship between the activation or suppression of caspase1, 3, 8, and 9 signaling [61–65]. While caspase activation is required for HIV-1 release in vitro, it is unclear whether caspase activation also controls viral release from productive infected CD4+ T cells in vivo and from HIV reservoirs. To investigate if HIV-1 release in infected individuals also requires caspase activation, we isolated CD4+ T cells from two chronically HIV-infected viremic individuals with plasma viremia between $10^4–10^5$ copies/ml of HIV RNA and stimulated them with anti-CD3 antibody for viral release in the presence of Belnacasan, ZVAD-FMK, QVD-OPH or DMSO. The released cell-free virions in the culture supernatants were quantified on day 3 and 6 using the COBAS Ampliprep/COBAS TaqMan HIV-1 test. While Belnacasan and ZVAD-FMK inconsistently inhibited the viral release, QVD-OPH consistently

suppressed viral release on both time points with up to 10,000-fold reduction in virion-associated HIV RNA (Fig 7A). To assess if HIV-1 release from viral reservoirs also requires caspase activation, we isolated CD4+ T cells from three aviremic HIV-infected individuals receiving ART and stimulated the cells with anti-CD3 antibody for 4 days in the presence of QVD-OPH or DMSO. The presence of QVD-OPH reduced the viral release from the CD4+ T cells of the aviremic individuals (Fig 7B). To further address a potential inhibitory effect of QVD-OPH to CD3 stimulation, we performed the viral release experiment using CD4+ T cells derived from three viremic individuals in the absence of CD3 stimulation. While much less viral loads were detected without CD3 stimulation, QVD-OPH suppressed viral release in cells derived from all three individuals (Fig 7C). Together, caspase inhibition significantly suppressed viral release from CD4+ T cells of infected individuals (Fig 7D), suggesting that HIV-1 releases from both chronically viremic and aviremic individuals are not spontaneous but rather requires caspase activation.

## Discussion

Host genes, such as tetherin (BST2) and TIM-family proteins, are known restriction factors of HIV release [12, 13]. To counter host release restriction, HIV uses its accessary proteins, such as Nef and Vpu, to induce the internalization and degradation of the host restriction factors [14, 66]. While the restriction factors are part of host antiviral response, it is not known that any host genes other than transcriptional and translational machinery may facilitate HIV release. Here we report that HIV-1 infection activates the cellular caspase pathway to induce PS exchange and the cleavage of adhesion receptor CD62L to facilitate the viral release (Fig 8). The requirement of caspase activation in HIV-1 release demonstrated, for the first time, that HIV-1 viral release is not a spontaneous process but rather is dependent on the inflammatory apoptosis in the infected cells, and host cell caspase activation becomes an integral part of the viral release phase of HIV-1 life cycle. Our current and previous findings support a dual functionality of CD62L in HIV-1 infection. While the binding of HIV envelope glycan to the C-type lectin receptor CD62L facilitates the viral attachment and entry to CD4 T cells [15], the same glycan-CD62L interaction may hinder viral release by tethering virions on the surface of budding T cells (Fig 5F). Cleavage of CD62L effectively prevents the selectin-mediated tethering of releasing virions. As CD62L shedding is controlled by inflammatory ADAM metalloproteinases, whose activations require caspases-induced PS-flipping, HIV viruses are evolved to express Nef to induce host cell apoptotic caspase activation, leading to the enzymatic removal of the viral attachment receptor by ADAM metalloproteinases.

The current paradigm assumes HIV-1 infection-induced caspases activation is part of host anti-viral response to kill infected T cells [8], thus detrimental to the viral expansion. This prompted a treatment strategy to stimulate procaspase8 expression and to enhance apoptosis of latency reversal agents (LRA-kick) reactivated viral reservoir cells, as some of the stimulated viral reservoir cells resisted apoptotic cell death [67]. Our findings of viral release required caspase activation suggests, however, that caspases activation is not detrimental but potentially beneficial to the viral expansion. This shift in the role of caspases in HIV infection highlights a potential risk of stimulating caspases as a treatment strategy, that may lead to production of more viral progeny. We think the amount of released virus may outcompete the benefit of causing reservoir cell death. If the death of stimulated viral reservoir cells preceded viral release (namely the benefit in viral release do not outcompete the death of reservoir cells), then inhibition of caspases should lead to more but not less released viruses. Our results of ex vivo stimulated HIV-1 release from viral reservoir cells showed caspase inhibition in these experiments suppressed viral release (Fig 7), suggesting that viral release likely preceded cell death.

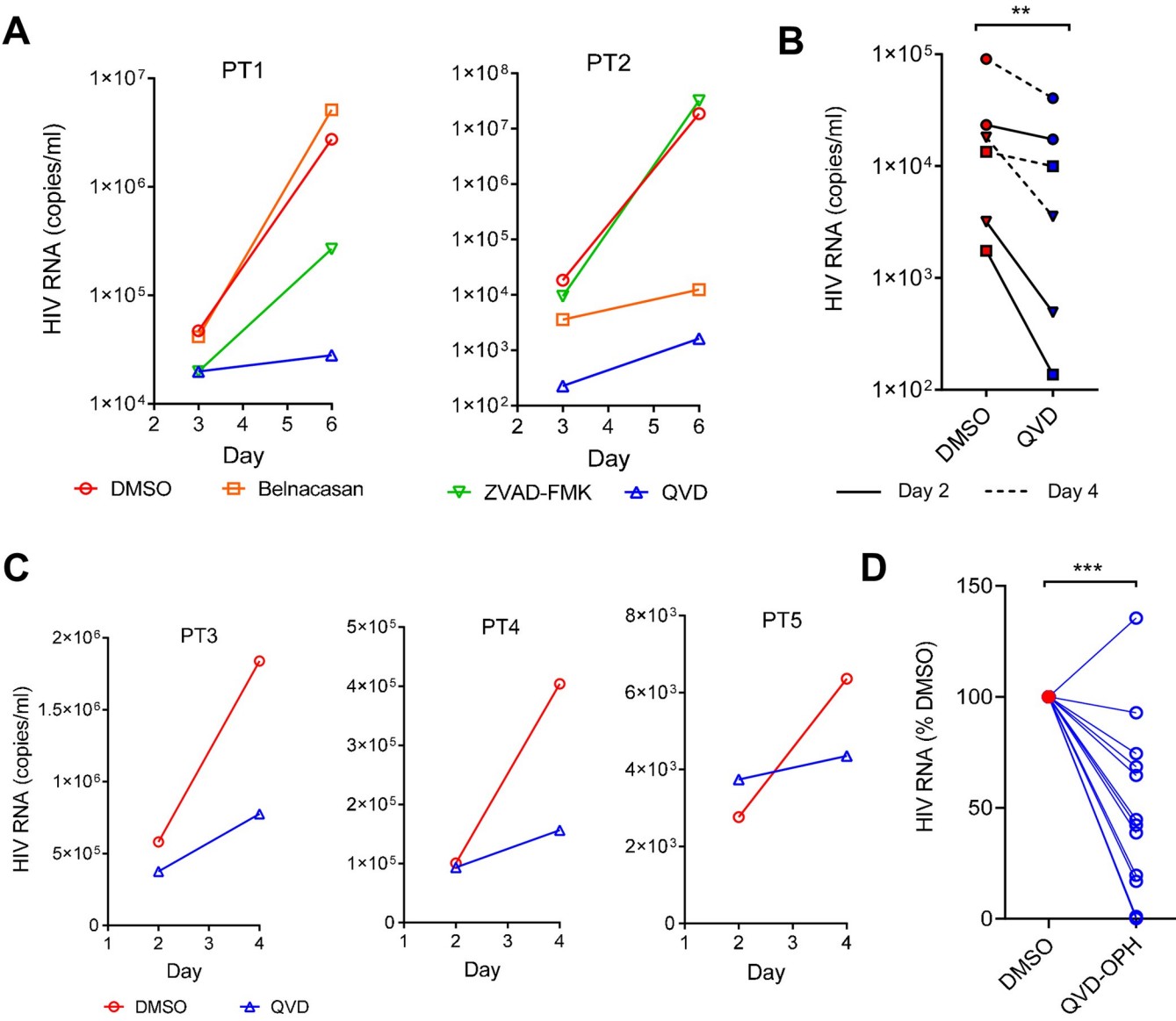

**Fig 7. Caspase Inhibitor suppresses the ex vivo viral reservoir. (A)** HIV viral release was measured as HIV RNA copies/ml in the presence of various caspases inhibitors or DMSO from CD4 T cells derived from infected individuals but stimulated *in vitro* with anti-CD3 for 3 and 6 days. PT1 and PT2 are two infected viremic individuals not receiving ART treatment. **(B)** Effect of QVD-OPH in suppressing HIV release from viral reservoirs. Viral RNA (copies/ml) were quantified from CD4 T cells in day 2 and 4 of anti-CD3 stimulated culture supernatants in the presence of QVD-OPH or DMSO from three aviremic individuals on ART treatment. **(C)** QVD-OPH inhibited HIV viral release from non-stimulated CD4 T cells derived from three infected individuals. **(D)** Percentage of HIV released from CD4 T cells with QVD treatment compared to that of DMSO from all infected viremic and aviremic individuals. The p-value is calculated using ordinary two-way ANOVA between QVD and DMSO treated samples.

Despite their hierarchal activation, only pan-caspase inhibitors effectively suppressed experimental HIV-1 infection and viral release. Inhibitors of individual caspases and their combinations failed to inhibit viral infections, suggesting the presence of redundancy among caspases and their activation hierarchy is less relevant in HIV-1 viral release. The role of caspases in viral release may be different from viral replication, in which caspase 8 appears critical [11]. This is consistent with the notion that PS exchange occurs from several activated caspases and perhaps why targeting of individual caspases is not sufficient despite the involvement of individual caspases during acute infection [68–70]. It is interesting to note that TIM-family

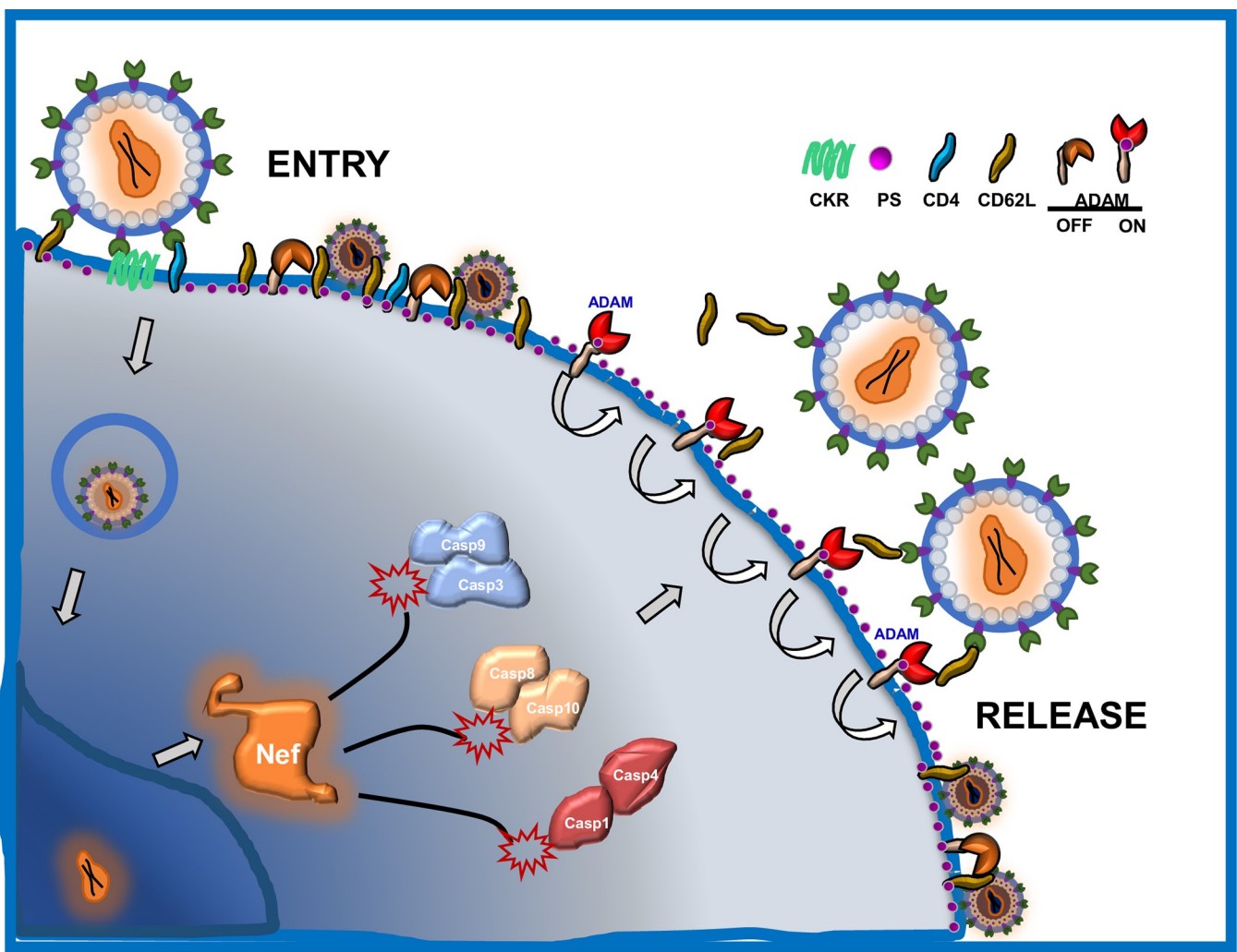

**Fig 8. Targeting caspase activation suppresses the viral reservoir.** Model of Nef-mediated activation of caspases for promoting PS exchange thus stabilizing sheddase activity to cleave CD62L and permit the release of viral progeny. Inhibition of this activation causes downstream tethering of budding virus by CD62L resulting in a phenotypic change of viral morphology and loss of fitness.

proteins restrict HIV release via PS-dependent mechanism [14]. It is possible that the TIM mucin-domain binding to PS may interfere with caspase-induced PS activation of ADAM shedding metalloproteinases, and thus restricting the viral release.

Caspase inhibition resulted in the emergence of potentially diminutive viral progeny as well as more virions attached to CD62L (Fig 5). It is interesting to note that while the number of mature virions decreased in response to caspase inhibitions, the number of smaller potentially virion-like particles (VLP) increased (Fig 5D), and they appeared to be deficient in infection (Fig 5E). It is worth noting that pan-caspase inhibitors may cause necroptosis and other forms of cell death, resulting in excessive cellular debris that resemble smaller virions. This is unlikely, however, as the concentration of QVD-OPH (25 μM) in these experiments did not show excessive cell death compared to the control treatments (S4 Fig). Targeting of an upstream initiator with a more efficacious agent supports our previous findings with enzymatic inhibition of CD62L cleavage by an ADAM metalloproteinase inhibitor. It remains to be seen if targeting caspase-mediated viral release can effectively reduce viral reservoirs in vivo,

either alone or in combination with ART, and in patients with multidrug resistant strains of HIV-1 [71]. As caspase inhibition targets a unique step in HIV life cycle different from those of current ART regimes, it is conceivable that pan-caspase inhibition may synergize with existing ART to further reduce viral load in infected individuals. Recently, QVD-OPH was shown to prevent AIDS progression in SIV-infected rhesus macaques [72], further supporting of targeting caspases as a potential anti-HIV measure.

Previous findings implicated Nef in many inflammatory pathways, such as promoting TNFα secretion through ADAM17 and ADAM10 [26, 49, 73, 74], as well as downregulating cell-surface receptors [75–77]. Indeed, investigations into various nef gene contributions showed virulence, or the viral capability to expand upon founding infection, coincided with the ability to activate caspases. For infections involving nef-null (ΔNef) or HIV-2 (BEN), limited viral expansion appeared alongside limited caspase activation whereas HIV-1 wild type or SIV variants (mac239) or (SMM) expand and activate caspases readily. Interestingly, an HIV-1 variant from an asymptomatic individual (NA7) showed limited caspase activation and viral expansion compared to the wild type. While Nef is important for caspase activation, it is likely that other viral genes, such as Vpu, also contribute to caspase activation since QVD-OPH suppressed residual caspase activation from nef-null HIV-1 infection (Fig 6D).

Early work had shown HIV-1 infection induced pathogenic CD4 T cell death, a hallmark during the progression to AIDS in HIV and SIV [78]. While unclear of its mechanism, several groups have hypothesized that the viral-induced T cell apoptosis could reduce immune function against the viral infection and suggested to use anti-apoptotic measures to counter HIV pathogenesis [51, 79, 80]. Indeed, the apoptosis of by standard T cells was further established as evidence of viral down-modulation of immune response [4]. Despite the inhibition of caspase activation may increase viral replications in infected cells, less infectious virus was released. Since the net outcome of inhibiting caspases is a reduction in viral transmission, we conclude that apoptotic caspase activation serves to benefit the viral pathogenesis by facilitating the viral release.

The evolution of lentiviruses has been characterized as rapidly diversifying, however, the known coexistence in homo sapiens is appreciably early compared to other viruses [81]. Our finding of attachment receptor (CD62L) cleavage-dependent HIV-1 release may be a more general theme characterizing a non-spontaneous viral release that requires shedding of attachment receptors. Indeed, a similar mechanism exists for influenza virus, a member of the Orthomyxoviridae family of RNA viruses, in which neuraminidase cleavage of sialic acids is required for influenza viral release [82, 83]. Mutations inactivating viral neuraminidase resulted in aggregation of virions on budding cells, thus, deficiency in viral release [82]. However, unlike influenza virus which encodes neuraminidase in its genome, HIV-1 does not encode an attachment receptor cleavage enzyme. Instead, it relies on host metalloproteinases to remove CD62L. As the enzymatic activities of ADAMs are controlled by caspase-mediated inflammatory apoptosis, this obligates HIV-1 to use Nef to induce inflammatory response and caspase activation in host cells. Nef-deficient HIV-1 is therefore disadvantageous in its viral release. Consistently, HIV-1 budding virions in the presence of caspase inhibition exhibited phenotypical resemblance to budding influenza virus with deficient neuraminidase.

While tetherin and TIM-family restriction factors are important in host antiviral responses, their therapeutic potentials are limited. In contrast, caspases can be targeted with therapeutic compounds. Caspase inhibition suppressed viral release from HIV-1 reservoirs of aviremic and viremic infected individuals. Thus, HIV-1 release from cellular reservoirs also requires caspase activation. While current antiviral therapies are based on stimulating apoptotic cell death to eliminate infected CD4+ T cells [9], our findings suggest this approach may not be effective as HIV alone stimulated infected cell death to facilitate its release. Instead, we suggest

a potential new anti-HIV therapy based on inhibitors targeted at inflammatory viral release, hence "plugging the viral reservoirs".

## Conclusions

HIV infection remains incurable to date and there are no compounds targeted at the viral release. We show here HIV viral release is not spontaneous, rather requires caspases activation. Blocking the caspases activation caused virion tethering by CD62L and the release of deficient viruses. Not only productive experimental HIV infections require caspases activation for viral release, HIV release from both viremic and aviremic patient-derived CD4 T cells also require caspase activation, suggesting the release of viruses from their cellular reservoirs depends caspase activation. Transcriptomic analysis of HIV infected CD4 T cells showed a direct contribution of HIV Nef to apoptotic caspases activation. Current HIV cure strategy seeks to stimulate T cell apoptosis to eliminate latently infected cellular HIV reservoirs. Our work has shifted the paradigm on HIV-induced apoptosis and suggests the current approach may induce HIV release and thus be counter-productive. Instead, our study supports targeting of viral reservoir release by inhibiting of caspases activation.

## Materials and methods

### Reagents

Unless otherwise specified, all reagents and chemicals were purchased from Sigma-Aldrich Co. (St. Louis, MO). Anti-human CD3 antibody (OKT3) was kindly provide by Dr. Gilliland of Janssen and used for stimulation of all primary PBMC. Fluorescently labeled antibodies for flow cytometry against: CD8, CD4, CD3, CD62L, annexin V, and their isotype controls (IgG1, IgG2A, IgG2B) were obtained from BD Biosciences (San Jose, CA), BioLegend (San Diego, CA) or eBioscience (San Diego, CA). APC-labeled CD62L antibodies and BV421-labeled annexin V antibodies used for confocal microscopy were obtained from BioLegend (San Diego, CA) and BD Biosciences (San Jose, CA), respectively. HIV-1 core antigen antibody (KC57-FITC/PE) for intracellular p24 staining was purchased from Beckman Coulter, Inc. (Miami, FL). Recombinant IL-2, also known as Teceleukin or Tecin (RO23-6019, Roche, MN) was obtained from National Cancer Institute, NIH. The Luciferase Assay System was purchased from Promega Corporation (Madison, WI). HIV-1 p24 ELISA kit was obtained from PerkinElmer Life Sciences, Inc. (Waltham, MA). Ficoll-Paque was purchased from GE Healthcare Life Sciences (Pittsburgh, PA). The Vybrant FAM poly-caspase assay from Molecular Probes-Thermofisher (Waltham, MA) was used for intracellular, active-caspase staining of pan-caspases. For individual caspase expression measurements, caspase-1, -2, -3/7, -6, -8, -9, and -10 fluorescent inhibitors of caspases (FLICA) were used to stain active-caspases in live cells from Bio-Rad (Hercules, CA). DMEM, RPMI1640, penicillin/streptomycin (Pen/Strep), fetal bovine serum (FBS), and HEPES were purchased from Invitrogen Corporation (Carlsbad, CA). The metalloproteinase inhibitor, BB-94 (Batimastat), TAPI-1 (TNFα protease inhibitor-1) and TAPI-2 were purchased from Santa Cruz Biotechnology. Caspase inhibitors Z-VDVAD-FMK (caspase 2), and Z-DEVD-FMK (caspases 3,6,7, and 10) were purchased from Santa Cruz Biotechnology, Z-DQMD-FMK (caspase 3), Z-IETD-FMK (caspase 8) and Z-LEHD-FMK (caspase 9) were purchased from Adooq, Ac-DEVD-CHO (caspase 3, 7) and Ac-LEVD-CHO (caspase 4) were purchased from Sigma-Aldrich, Belnacasan (caspase 1 and 4) and pan-caspase inhibitors Z-VAD-FMK and Q-VD-OPH were purchased from MedKoo Biosicences (Morrisville, NC) or SM Biochemicals LLC (Anaheim, CA).

## Preparation of pseudotyped and replication competent virus

The use of human peripheral blood mononuclear cells (PBMC) is approved by the Department of Transfusion Medicine at the Clinical Center of National Institutes of Health. PBMC's were isolated by Ficoll-Paque gradient from randomly selected, non-identified healthy donors. The isolated PBMC were distributed at $3x10^6$/mL in 12-well plates with RPMI supplemented with 10% FBS, 1% Penicillin/streptomycin and 20U/mL IL-2, activated with 1–2μg/ml anti-CD3 antibody Okt3 for 48 hours. Prior to all infections, CD8$^+$ cell depletion was completed using the EasySep Human CD8 Positive Selection Kit (Vancouver, BC, Canada). Total cell count and viability determinations were assessed with the Muse Cell Analyzer System (Millipore) or the Luna FL Dual Fluorescence Cell Counter (Logos Biosystems); the viability of cells under various compound treatment were comparable to those of DMSO. All data presented in this manuscript were results of replicates of individual typical experiments derived from at least two independent experiments/donors. The expression vector for pNL4-3.Luc.R-E- was obtained through the NIH AIDS Research and Reference Reagent Program [84]. The HIV viral vector, pNL4-3.Luc.R-E-, which contains the firefly luciferase gene inserted into the HIV-1$_{NL4-3}$ *nef* gene and frameshift mutations to render it E-, was used to generate all pseudotyped viruses [85]. Recombinant HIV luciferase viruses for both R5 and X4-tropic pseudovirus strains were generated by co-transfecting 293T cells with 5 mg of the HIV-1$_{NL4-3}$ backbone and 5 mg of either of the HIV envelopes, as previously described [86]. Virus collected in the culture supernatant were quantified by HIV p24 ELISA and adjusted to 1 mg/ml p24.

HIV-1 M-group CCR5-tropic Ba-L (HIV-1$_{BAL}$) and CXCR4-tropic BRU (HIV-1$_{LAI}$) strains were obtained from the NIH AIDS Research and Reference Reagent Program (https://www.aidsreagent.org/) under catalog numbers 510 and 2522, respectively. Both viruses were propagated by infecting stimulated, CD8-depleted PBMC's cultured in RPMI1640 (10% FBS, 1% Pen/Strep, 1% HEPES, 20 units/ml IL-2). Day 6 supernatant was harvested and frozen at -80°C in 200 μl aliquots. Viral titers in TCID$_{50}$ were determined by titrating viral infection in stimulated, CD8-depleted PBMC's and measuring p24 by ELISA using PerkinElmer ALLIANCE HIV-I p24 ELISA kit (PerkinElmer Life Sciences, Inc., Waltham, MA). HIV-1 pNL4-3 proviral vectors incorporating the wild-type sequence encoding Nef as pBR43IeG-nef+ (Nef+) or possessing mutations disrupting the ORF of Nef as pBR43IeG-nef- (Nef-) as well as NL4-3 Nef variants including HIV-2 (BEN), HIV-1 Group M (NA7), and SIV (mac239), TAN1 (CPZ), FYr1 (SMM) were obtained from the NIH AIDS Research and Reference Reagent Program (https://www.aidsreagent.org/) under the catalog numbers 11349,11351, M30502, DQ242535, M33262, AF447763, and DQ092760, respectively. The plasmids were first transfected into 293T cells plated in 6-well plates at $5x10^5$ cells/well 24 hours prior in DMEM media (10% FBS, 1% Pen/Strep) using the calcium phosphate transfection kit from Invitrogen (ThermoFisher Scientific, Waltham, MA). Media was changed 24 hours later to RPMI1640 (10% FBS, 1% Pen/Strep). The Nef variant NL4-3 clones were amplified by infecting CEM T cells. Media containing virus was harvested 48 hours later and stored at -80°C in 500 μl aliquots. Virus was quantified by ELISA for p24 using PerkinElmer ALLIANCE HIV-1 p24 ELISA kit (PerkinElmer Life Sciences, Inc., Waltham, MA) for infections that were carried out using equal amounts of virus for each clone.

## Infections with pseudovirus and replication-competent virus

HIV infection here includes viral entry, replication, spread and cell-cell transmission in replication-competent viruses. It reflects viral entry in pseudovirus experiments. Stimulated, CD8-depleted PBMC's were resuspended at $2x10^6$/ml in culture media. Aliquots of 200 μl ($4x10^5$ cells) were transferred to 96-well plates, and pseudotyped HIV-1 JRFL (R5-tropic) and

SF33 (X4-tropic), or VSV were added to the cells at a concentration of 100 ng/ml HIV p24 as previously reported [15]. The infected PBMC's were then incubated at 37˚C for 72 hours, lysed, and assayed for luciferase activity according to manufacturer's recommendations (Promega Corporation; Madison, WI) using a Tecan X-10M Luminometer (Tecan Life Sciences, Switzerland).

For infection with HIV-1$_{BAL}$, HIV-1$_{LAI}$, or Nef variant HIV-1$_{NL4-3}$ virus strains, CD8-depleted PBMC cells were resuspended at $2x10^6$/ml in RPMI1640 (10% FBS, 1% Pen/Strep, 1% HEPES, 20 units/ml IL-2). Cells were incubated with virus based on ~$4x10^5$ TCID$_{50}$ or standardized p24 amount, respectively, for 1 hour before being washed and suspended in new media with or without inhibitor compounds and incubated in a 24-well plate along with uninfected controls at 37˚C with 5.5% CO$_2$. Concentrations of caspase inhibitors used were: 100 µM for all screening assays and 50 µM in all assays directly comparing pan-caspase inhibitors QVD-OPH and ZVAD-FMK to Belnacasan and BB-94 controls, and 25 µM for electron microscopy analyses. The concentration of nelfinavir used is 40nM in infections. Cell cultures and their supernatants were harvested on day 6, or as otherwise indicated following infection. Infections were detected by either intracellular p24 at indicated time points or RT-PCR for viral RNA copy number at day 3 following infection. Intracellular p24 levels were measured on viable CD3$^+$ populations using FITC-conjugated KC57 antibody using the Cytofix/Cytoperm kit from BD Biosciences (San Jose, CA). Samples were collected on a FACSCanto II (BD Biosciences). For infections with replication-competent VSV (Indiana strain), $2x10^6$ stimulated cells/ml of PBMC's were infected 1:5,000 (v/v) with diluted virus for 1 hour before being washed and cultured 48 hours in the presence or absence of 50 µM QVD-OPH. Supernatant media containing virus was harvested and applied onto Vero cells for plaque assays.

RT-PCR using HIV-1 LTR or caspase primer probes was assayed as described previously [15, 87] using TaqMan primer/probe sets from ThermoFisher (Waltham, MA) with a 50 ng cDNA or gDNA input. The following TaqMan 20X Gene Expression Assay probes (Dye: FAM-MGB) were used: Caspase 1 (Hs00354836_m1), Caspase 3 (Hs00234387_m1), Caspase 8 (Hs01018151_m1), and HIV-1 LTR (Pa03453409_s1). Expression Assay standard probes (Dye: VIC-MGB) ACTB (Hs01060665_g1) or RNase P/RPPH1 (4403328) were used as gene controls. Samples were run in duplicate across triplicate experiments on an ABI 7300 Real-Time PCR System (Applied Biosystems) with a threshold of 0.2. Relative gene expression was calculated using the $2^{-\Delta Ct}$ method and normalized to β-Actin or RNaseP as the endogenous controls. Viral replication following integration was determined using the same probes. Briefly, infected cells from triplicate experiments were washed 3x in PBS and divided for gDNA and RNA isolation separately. Replication efficiency was measured from cDNA derived from RNA following normalization of integrated viral copies against ACH-2 standard with a single integrated copy.

## Caspase-annexin multi-staining of HIV-1 infected cells

Primary PBMC's were infected for 6 days in the presence or absence of QVD-OPH or Belnacasan with HIV-1$_{BAL}$ or nef$^+$ and nef$^-$ HIV-1$_{NL4-3}$ before being harvested. Intracellular p24 staining requires permeabilization whereas accurate measurement of PS exchange depends on membrane integrity thus are not compatible staining assays, hence cells for each condition were equally divided. Briefly, cells were gently centrifuged at 500 g for 5 mins and 100 µl fresh media was added to each for staining for activated poly-caspases per manufacturers protocol for 1 hour at 37˚C. For cells labeled with intracellular p24, samples were washed 2x with PBS before permeabilization and fixation for 1 hour as previously described before being washed and labeled for p24, CD62L, CD4, and CD3. For cells labeled with annexin V, samples were

washed 2x with PBS before being labeled for surface CD62L, CD4, and CD3 for 1 hour at 4˚C. Samples were washed in PBS and suspended in apoptosis binding buffer and labeled for annexin V for 15 mins at room temperature per manufacturers protocol. Samples were washed in apoptosis binding buffer before all samples were collected on a FACSCanto II.

## Immunoprecipitation and immunoblot assays

CD8-depleted PBMC's were prepared using magnetic bead EasyStep human CD8 positive selection kits II (Cat 17853 StemCell Technologies) before being infected with HIV-1$_{BAL}$ (NIH HIV Reagent Program) using a predetermined TCID50 as previously described. A total of $40 \times 10^6$ cells were used per condition in the presence of 50 μM QVD-OPH, 40 nM Nelfinavir, or DMSO. Media was changed and fresh compounds added accordingly on Day 3. On Day 6, infected cells were harvested, washed in 1x PBS, and lysed in SDS-free Pierce IP lysis buffer (Cat 87787 ThermoFisher). Aliquots of heterogenous, whole-cell lysates were taken for loading controls. HIV gag proteins were precipitated out using the Dynabeads Protein G immunoprecipitation kit (Cat 10007D ThermoFisher) labeled with unconjugated KC57 coulter clone (IgG1 FH190) antibody (Beckman Coulter) before undergoing SDS-PAGE in 10–20% Novex Wedge-Well tris-glycine gels (Cat XP04200BOX ThermoFisher). Proteins were transferred to 0.45μM nitrocellulose membranes using the iBlot2 dry blotting system (ThermoFisher) before being probed with primary rabbit anti-p55/p24/p17 polyclonal antibody (Cat ab63917 Abcam) or IgG2b mouse anti-human β-actin monoclonal clone BA3R antibody (Cat MA5-15739 ThermoFisher) at 1:2,000 in 2% BSA PBS-T (1x PBS/0.1% Tween20) for 1 hour. Membranes were washed in PBS-T for 5 mins before being labeled with secondary goat IgG polyclonal anti-rabbit Fc-HRP (Cat 31463 ThermoFisher) or goat IgG (H+L) SuperClonal recombinant polyclonal anti-mouse-HRP antibody (Cat A28177 ThermoFisher) at 1:2,000 dilution for 1 hour. Membranes were washed in PBS-T before being developed using SuperSignal West Dura Extended Substrate Kit (Cat 34076 ThermoFisher) and analyzed using the FluorChem M imaging platform (ProteinSimple).

## Trypsin-mediated viral release assay

PBMC's were infected as previously described in the presence and absence of Belnacasan, QVD-OPH, or BB-94 for 6 days. To determine the distribution of viral progeny in the presence of these compounds, supernatants containing cell-free virions were collected, and the infected cells were washed with 1x PBS before being treated with 1x cell culture trypsin-EDTA solution from Gibco-Invitrogen (ThermoFisher, Waltham, MA) or media for 15 minutes at 37˚C. Trypsin was then inactivated 1:1 (v/v) with media supplemented with 20% FBS. Free virus or membrane-associated virus collected from media or by trypsinization, respectively, was applied on TZM-BL reporter cells seeded 72 hours prior at 5,000 cells/well in a 96-well plate and incubated for 72 hours at 37˚C. Samples were washed and lysed as previously described [15]. Transduction of a luciferase signal as a function of viral concentration was measured using a Tecan X-10M Luminometer (Tecan Life Sciences, Switzerland).

## Cell to cell transmission of virus

For the cell-cell transfer-mediated infection, TZM-BL cells were seeded in a 96-well, flat-bottom plate at 5,000 cells/well three days before the assay. PBMC's infected with HIV-1$_{BAL}$ for 6 days with or without the presence inhibitor compounds were harvested and washed in 1x PBS to remove free virus before being suspended in fresh media and added to the wells at a PBMC: TZM-BL concentration ratio of 4:1, 2:1, 1:1, and 1:2 cells/well in equal volume. Fresh inhibitor compounds were added to samples that received corresponding treatment for the cell-cell

transfer assay. All conditions and dilutions were prepared in triplicate. The cell mixtures were incubated at 37°C for 72 hours, followed by lysis and measurement of the subsequent luciferase expression as previously described.

## Viral fitness assay

Stimulated PBMC's were infected at $2 \times 10^6$ cells/ml confluence with 20, 10, 1, 0.5, and 0.1 ng dilutions of HIV-1$_{BAL}$ by p24 concentration in triplicate and incubated for 6 days in the presence of 50 μM QVD-OPH, BB-94, or DMSO. To determine infectivity, cells were harvested and centrifuged at 1200 RPM for 5 mins. Media was recovered and serially diluted before applying onto target TZM-BL reporter cells for each p24 concentration. To determine HIV p24 capsid produced, aliquots from the same recovered media were assayed for p24 concentration by ELISA. Viral fitness was determined by normalizing all conditions by p24 concentration before comparing infectivity.

## HIV-1 release from patient-derived CD4+ T cells

PBMC's from HIV-1 infected individuals were obtained by leukapheresis and ficoll-hypaque centrifugation. The use of PBMC from HIV-1 infected individuals for the study was approved under protocol #02-I-0202 by the Institutional Review Board (IRB) of National Institutes of Health with written informed consent from all donors. CD4+ T cells were isolated using a cell separation system (StemCell Technologies). Cells were cultured with plate-bound anti-CD3 and soluble anti-CD28 antibody in the presence of various compounds in duplicate for 48 hours. The copy number of virion-associated HIV RNA released to the cell culture supernatants was determined using the Cobas Ampliprep/Cobas Taqman HIV-1 Test, Version 2.0 (Roche Diagnostics). The limit of detection was 20 copies/ml.

## Transcriptome analyses through next generation RNA sequencing

For all sequencing samples CD8-depleted PBMC's were stimulated 48 hours before infection with HIV-1$_{BAL}$ and plated at a confluency of $2 \times 10^6$ cells/ml. For sequencing involving PBMC's, samples were harvested, and RNA extracted. For sequencing involving CD4+ T cells only, CD4+ T cells were further isolated from PBMC's using EasySep human CD4+ T cell enrichment kit (STEMCELL Technologies Inc, MA 02142) prior to RNA extraction. To prepare samples for RNA sequencing, $2 \times 10^6$ PBMC's or purified CD4+ T cells from either infected or uninfected samples were 3x washed in 10 ml of 1x PBS to remove excess virus before being lysed for RNA isolation using RNeasy mini kit (QIAGEN). Samples had genomic DNA further removed by treatment using DNase Max kit (QIAGEN). Samples of 2–10 ug RNA with concentration >25ng/ul and OD260/280 >2.0 were used for cDNA library preparation of 250–300 base pair fragments in length, and high throughput sequencing using Illumina Mi-Seq platform by Novogene (Novogene Corp Inc, CA). A total of ~$20 \times 10^6$ reads were recorded for each sample (S1 Table). The mapping and assembly of raw sequencing reads, statistical and preliminary bioinformatic analyses were performed by Novogene. Subsequent gene expression analyses were performed based on expected Fragments Per Kilobase of transcript sequence per Millions base pairs sequenced (FPKM) using sequences of 10 reads or higher. All differential gene expression analyses were done between infected samples in the presence or absence of treatment and their donor matched uninfected controls for both PBMC and CD4+ purified sample sets. The values of differential gene expression are calculated as fold of gene expression difference between the infected (with or without the treatment) and uninfected samples over the average between infected (no treatment) and uninfected samples. To avoid differences associated with donor variations, all comparisons are done for samples within each donor. The differential gene expressions between HIV-1$_{BAL}$ and uninfected

range from 2 to -2, representing complete upregulation and complete downregulation of genes, respectively.

## Enzymatic cleavage assays

Cell membrane-associated proteins were extracted from CD8 depleted PBMC using a plasma membrane protein extraction protocol as previously described [88]. Approximate $2.0 \times 10^7$ cells were spun down and resuspended in 2 ml ice-cold lysis buffer containing 50mM HEPES, pH 7.4, 150mM NaCl, and 40μg/ml digitonin. The cell suspension was frozen for 1 hour to promote lysis. After freezing, the cells were thawed and spun down in micro-centrifuge. The cell pellet was washed once with 2ml of 50 mM HEPES, pH 7.4, 150 mM NaCl. The membrane-associated proteins were extracted using 50mM HEPES, pH 7.4, 150mM NaCl buffer containing 1% Nonidet P40, and dialyzed against 50% ADAM enzymatic assay buffer (2.5 μM $ZnCl_2$, 0.005% Brij-35, 25 mM Tris, pH9.0). The enzymatic cleavage of L-selectin/CD62L was carried out using a fluorescence-based enzymatic assay. The fluorogenic substrate was a Dabcyl (4-(4-dimethylaminophenylazo)benzoyl)- and Edans (5-[(2-aminoethyl)amino]naphthalene-1-sulfonic acid)- conjugated 12 amino acid peptide, Dabcyl-KLDKSFSMIKEG-Edans, synthesized by Biomatik (www.biomatik. com). The substrate peptide corresponds to the extracellular membrane proximal region of human L-selectin encompassing the identified ADAM cleavage site [89]. The fluorescence of the Edans group is quenched by the Dabcyl group in the intact substrate and the cleavage of the substrate peptide results in a fluorescence emission at 490 nm. The cleavage assay was carried out at 37°C for 4 hours in a 96-well plate using Synergy-H1 fluorescent plate reader (BioTek Instruments, Inc. VT) with excitation and emission wavelengths of 336 nm and 490 nm, respectively. Each enzymatic reaction contained 1ul of membrane-associated ADAM extract mixed with 5–10 uM fluorescent peptide with or without indicated inhibitors in 100 μl ADAM assay buffer (25 mM Tris at pH 9.0, 2.5 μM $ZnCl_2$, and 0.005% Brij-35 (w/v)). For on-cell cleavage of the fluorogenic CD62L-peptide substrate, $2 \times 10^6$ infected or uninfected CD8-depleted PBMC were centrifuged at 300g for 5min to remove culture media. The cells were washed once with 2ml fresh culture media (RPMI 1640 with 10% FBS, 1% Pen/Strep), and centrifuged again. After wash, the cells were resuspended in 1ml culture media containing 10 μM fluorescent substrate peptide in the presence or absence of indicated compounds and incubated at 37°C for 4 hours. 150 μl of the digestion supernatant was harvested and diluted 1:1 with lysis buffer and transferred to assay wells in a 96-well plate for fluorescence measurement using excitation and emission wavelengths of 336 nm and 490 nm, respectively. To measure soluble CD62L present in the infected and uninfected samples, cell culture media were refreshed 24-hours prior to the harvest of supernatants on day 6 of post infections. The concentrations of soluble L-selectin in supernatants were quantified using the human L-selectin DuoSet ELISA kit (R&D Systems) following the manufacture's protocol.

The enzymatic cleavage of HIV protease substrate by recombinant HIV-1 protease was carried out in 96-well black bottom plates. The recombinant HIV-1 protease (Catalog SRP2152) and the fluorogenic peptide substrate (Catalog H6660) were purchased from SigmaAldrich. The assay was performed according to the manufacture protocol. In brief, 100ng of recombinant HIV protease was mixed with 5 μM fluorogenic substrate peptide in the presence of specified inhibitors in 100 μL of assay buffer (0.1M sodium acetate, pH 4.7, 1 M sodium chloride, 1 mM EDTA, 1mM DTT, 10% DMSO). The assay was read every 30 seconds with excitation and emission wavelengths of 340nm and 490nm, respectively.

## Confocal microscopy

PBMC's were stimulated with anti-CD3 antibody for 48 hours prior to incubation with 10 uM camptothecin in the presence or absence of compound inhibitors and incubated overnight at

37˚C in RPMI1640 (10% FBS, 1% Pen/Strep, 1% HEPES, 20 units/ml IL-2). CD4+ T cells were further isolated from samples using the StemCell EasySep Human CD4+ T Cell Enrichment Kit. Isolated CD4+ cells were washed in 1x PBS followed by labeling with APC-conjugated CD62L for 1 hour at 37˚C. Cells were washed 2x in PBS before suspension in annexin V binding buffer for apoptosis detection per manufacturers protocol with BV421-conjugated, recombinant annexin V for PS labeling. Cells were washed 2x in annexin V binding buffer, fixed with annexin V binding buffer supplemented with v/v 2% paraformaldehyde and plated in 100 µl at $1x10^6$ cells/ml on poly-l-lysine coated slides with coverslips and sealed for 24 hours with nail polish. Images were captured on a Zeiss LSM 880 AxioObserver with Airyscan confocal microscope. A 633 nm HeNe laser and 405 nm diode laser were used to excite bvCD62L and annexin V, respectively. All image post-processing performed with Zen Blue v 2.3.

## Transmission and scanning electron microscopy

Specimens for transmission electron microscopy (TEM) were fixed with 2.5% glutaraldehyde in 0.1 M Sorenson's buffer. Samples were post-fixed 1h with 0.5% osmium tetroxide/0.8% potassium ferricyanide, 1 hour with 1% tannic acid and overnight with 1% uranyl acetate at 4˚C. Samples were dehydrated with a graded ethanol series and embedded in Spurr's resin. Thin sections were cut with a Leica UCT ultramicrotome (Vienna, Austria) stained with 1% uranyl acetate and Reynold's lead citrate prior to viewing at 80 kV on a Hitachi 7500 transmission electron microscope (Hitachi-High Technologies, Tokyo, Japan). Digital images were acquired with an AMT digital camera system (AMT, Chazy, NY). For immunogold labeling, $\sim 10^6$ cells were washed with 2 ml PBS, gently centrifuged at 250 g for 5 minutes, resuspended in 100 µl of labeling buffer (PBS with 1% BSA), and incubated with 10 µg of mouse anti-human CD62L (clone FMC46, Thermofisher) for 1 hour on ice. After removal of the primary antibody and wash, cells were incubated with 10 µg of secondary goat anti-mouse IgG H&L conjugated to 10nm gold nanoparticles (Electron Microscopy Sciences) for 1 hour on ice, then washed and fixed in 2.5% glutaraldehyde with 0.1 M sodium cacodylate at pH 7.4.

For scanning electron microscopy (SEM), cells were adhered to silicon chips and fixed with 2.5% glutaraldehyde in 0.1 M Sorenson's buffer overnight at 4C. Specimens were post-fixed for 1 hour with 1% osmium tetroxide and dehydrated in a graded ethanol series. The samples were critical-point dried under $CO_2$ in a Bal-Tec Model CPD 030 dryer (Balzers, Liechtenstein), mounted on aluminum studs, and sputter coated with 75 angstroms of iridium in a model IBS/TM200S ion beam sputter coater (South Bay Technologies, San Clemente, California). Specimens were viewed at 5 kV in a Hitachi SU-8000 field emission SEM (Hitachi-High Technologies, Tokyo, Japan) using secondary imaging mode. Virion-like particles were identified in SEM images as 50–150 nm size cell surface associated spherical particles, excluding nodules appearing at the tip of filopodia. Virion-like particles in TEM were further characterized by the presence of capsid. The uninfected samples showed less than 10 virion-like particles per image using these criteria.

## Statistical analyses

All statistical analyses were carried out using Prism 8 software (GraphPad Software Inc). Unless otherwise specified, all data from experimental infections and corresponding analyses are representative of two or more independent experiments for reproducibility. *P* values determined using specified testing, assuming Gaussian distribution where normality testing allowed. *P* values <0.05 were considered significant. All graphs display means ± SD as error bars.

## Supporting information

**S1 Table. High-throughput RNA sequencing statistics from all donor infections.** Note: Samples labeled HIV, QVD, and UI are from experiments with HIV-$1_{BAL}$ whereas samples labeled Nef+ and Nef- represent experiments with HIV-$1_{NL4-3}$ with or without the gene sequence encoding functional accessory protein, Nef. Samples labeled 1, 2, or 3 correspond to donor. The number of total sequences represent all sequences with reads >0, and they include coding sequences as well as pseudogenes, processed transcripts, Long intergenic non-coding RNAs (LincRNA) and anti-sense sequences.
(PPTX)

**S1 Fig.** (A) Enzymatic cleavage of a fluorescent peptide substrate ES003 (R&D systems, Inc) or a fluorescent CD62L peptide by recombinant ADAM10 (R&D systems, Inc.). Each enzymatic reaction consists of 400 ng of recombinant ADAM10 mixed with 2 μM ES003 (left panel) or 10 μM CD62L (right panel) peptides in the presence or absence of 1 μM specified inhibitor compounds in 100 μl reaction buffer using a 96-well microtiter plate. (B) FACS analyses on the percentage of CD62L+ (left panel) and CD4+ (right panel) T cells in p24+ (red), p24- (green) infected or uninfected (black) samples. Caspase activation correlated with the loss of CD62L and CD4 expressions. The data is representative of three independent experiments. Mann-Whitney nonparametric test $^*p < 0.05$, $^{**}p < 0.01$, $^{***}p < 0.001$. (C) Volcano plot of fold change in gene expression between HIV-$1_{BAL}$ infected and uninfected CD4 T lymphocytes by day 6 (from representative data in Fig 1D). Genes including members from IFI, IL, CCL, CCR, CXCR, and CDC groups involved in signaling and cell cycle regulation with significant changes in expression are highlighted in red (up-regulated) or blue (down-regulated), whereas genes involved in transcriptional regulation are highlighted in green. Among the most significantly up-regulated genes in infected cells are several interleukins and their receptors, such as IL-1β, IL-8, IL-18R and IL-23R.
(PPTX)

**S2 Fig.** Differential expressions of members of Interferon-inducible (A), TRIM (B), Chemokine receptors (CCR) and SERINC (C) family genes by RNA-Seq analyses on CD4 T cells enriched from day 6 *in vitro* HIV-$1_{BAL}$ infected PBMC from 3 donors. Several members of interferon-inducible genes such as IFI16 transcriptional activator and antiviral sensory responders IFI44 and IFIT (interferon-inducible transmembrane protein) are upregulated in the presence of the infection (A). Several antiviral response factors such as the tripartite motif family members (TRIM) including TRIM5, 22, and 56 are upregulated in response to HIV-$1_{BAL}$ infection in CD4 T cells from all 3 donors (B). HIV-$1_{BAL}$ infection upregulated the expressions of restriction factors SERINC5, SAMHD, and tetherin (BST2) but downregulated expressions of chemokine receptors (C). **(D)** RT-PCR probe of day 3 HIV-$1_{BAL}$ infected PBMC for the expressions of inflammatory (caspase 1), executioner (caspase 3), and initiator (caspase 8) classes of caspases. The results are displayed relative to that of beta-actin control. The statistics are calculated using student-test $^{**}p < 0.01$, $^{***}p < 0.001$, $^{****}p < 0.0001$. **(E)** Representative of FACS plots from FLICA staining of individual activated caspases in primary lymphocytes on day 7 of post infection with HIV-$1_{BAL}$. CD3+ cells treated with DMSO or QVD-OPH were gated on p24 capsid (top) before observing expression levels for each caspase. Samples treated with DMSO with defined populations of p24+ (red boxes) vs p24- (black boxes) were gated separately (left columns) while QVD-OPH and uninfected controls were gated on their total CD3+ populations (right columns). The results are representative of at least two experiments. **(F)** Activation of caspases in panel E grouped by initiator (caspases 2,8,9 10), executioner (caspases 3.6,7), and inflammatory (caspase 1) classes. $^*p < 0.05$, $^{***}p < 0.0005$.
(PPTX)

**S3 Fig. (A)** Representative FACS contour plots of primary CD4+ T cells treated with 10 μM CPT in the presence or absence of Belnacasan (orange), QVD-OPH (blue) or DMSO controls. Annexin V+/caspase+ and annexin V+/CD62L- populations are highlighted in colored gates. QVD-OPH but not Belnacasan significantly reduced annexin V+/caspase+ and annexin V+/CD62L- populations. **(B)** Statistical quantification of annexin V+/CD62L- populations within specified gates in panel **A** from two independent experiments. Mann-Whitney non-parametric test $^*$p < 0.05. **(C)** Confocal microscopy representative of two independent experiments of CPT treated and untreated primary CD4+ T cells stained with annexin V (blue) and anti-CD62L (red). The majority of cells were stained with either anti-CD62L or Annexin V but not both. The lower bar diagram shows percentage of cells labeled with CD62L (red bars), PS (blue bars), or both (dual, black bars) in confocal images under CPT treatment in the presence and absence of QVD-OPH. The statistical analysis was done using 2-way ANOVA with Tukey's method $^*$p < 0.05, $^{**}$p < 0.01. **(D)** Bar diagram showing the inhibition of CASP1 activation by 50 μM Belnacasan (orange) versus DMSO control (red). Mann-Whitney nonparametric student t test $^*$p < 0.05. **(E)** Histograms for the activation of caspases, annexin V staining and expression of CD62L corresponding to the FACS analyses presented in Fig 2A and 2B.
(PPTX)

**S4 Fig. (A-B)** Effect of combination of two caspase inhibitors to HIV-1 infection. Stimulated, CD8-depleted PBMC were infected with HIV-1$_{BAL}$ in the presence of caspase inhibitors Belnacasan (caspase 1 and 4), Z-DEVD-FMK (caspase 3,6,7 and 10), the combination of the two compounds, QVD-OPH or control DMSO. All compounds were used at 50μM concentration in duplicates. The infections were analyzed on day 7 of post infection by FACS for the percentage of infected cells that lost CD62L (**A**) and for the infection level (**B**). **(C-D)** Dose-dependent inhibition of HIV-1$_{BAL}$ infection by QVD-OPH **(C)** or ZVAD-FMK **(D)**. **(E)** Viability of PBMC treated with titration concentrations of ZVAD-FMK and QVD-OPH in triplicates. **(F)** The expressions of CD4 and CD62L on anti-CD3 stimulated PBMC on Day 7 in the absence and presence of 50 μM QVD-OPH. Treatment of QVD-OPH did not affect the expression of CD4 and CD62L. **(G)** Effect of caspase inhibition to VSV infection. PBMC infected with replication-competent vesicular stomatitis virus (VSV Indiana strain) in the presence of caspase inhibitor QVD-OPH or control DMSO. Data from two independent experiments. **(H)** Effect of caspase inhibitor QVD-OPH to HIV protease enzymatic cleavage of a fluorescent Gag peptide. The cleavage reaction was carried out in the presence of 100nM viral protease inhibitor, Saquinavir, Nelfinavir, 10μM caspase inhibitor QVD-OPH or control DMSO. **(I)** Representative of one out of four infection experiments used for immunoblots (Fig 5I–5K). CD8-depleted PBMC's were infected with HIV-1$_{BAL}$ in the presence of DMSO (red), QVD-OPH (blue), Belnacasan (orange), or Nelfinavir (green), and analyzed for total infections (p24+) in CD3+ populations.
(PPTX)

**S5 Fig. (A)** Representative transmission-EM image of PBMC's infected with HIV-1$_{BAL}$ in the presence of Belnacasan which consistently failed to suppress infection in all assays. Scale bar = 200 nm. **(B)** Fraction of virions associated with gold-conjugated anti-CD62L in the presence of QVD-OPH or DMSO. The percentage is tallied against the total number of virions observed in individual TEM images. Student t-test $^*$p < 0.05. **(C-D)** Population of T cells stained double positive for caspase and annexin V **(C)** or double negative for CD4 and CD62L in Nef+, Nef- HIV$_{NL4-3}$ infected PBMC or uninfected samples **(D)**. The FACS analyses were performed on CD3 positive T cells. Student-test $^{**}$p < 0.01. **(E).** Nef sufficient (white bars) and deficient (ΔNef, grey bars) virus infection of CEM#2 cells in the absence and presence of

50μM QVD. (**F**) RNA-Seq-based differential gene expression analyses on primary CD4 T cells infected with Nef+ or Nef- HIV-1$_{NL4-3}$ from 2 donors for genes involved in TLR or IL-2 signaling pathways. (**G**) Protein-interaction network connecting various apoptotic pathways as depicted by Cytoscape (http://cytoscape.org) and String-Db (http://string-db.org). The shared components among the pathways are highlighted in red box.
(PPTX)

**S1 Raw images.**
(PDF)

## Acknowledgments

We thank Drs. Malcolm Sim and Jinghua Lu for their insightful suggestions throughout the project, Dr. Biao He for his help in part of the experiments, Dr. Javier Manzella-Lapeira for his help in confocal experiments. We would also like to thank Erin Huiting for her help in the release assays involving patient-derived cells, and Zachary Stotz for his help on the manuscript.

## Author Contributions

**Conceptualization:** Peter D. Sun.

**Data curation:** Jason Segura, Joanna Ireland, Zhongcheng Zou, Gwynne Roth, Julianna Buchwald, Thomas J. Shen, Elizabeth Fischer, Tae-Wook Chun.

**Formal analysis:** Joanna Ireland, Zhongcheng Zou, Gwynne Roth, Julianna Buchwald, Thomas J. Shen, Elizabeth Fischer, Tae-Wook Chun, Peter D. Sun.

**Funding acquisition:** Peter D. Sun.

**Investigation:** Jason Segura, Joanna Ireland.

**Methodology:** Jason Segura, Joanna Ireland, Susan Moir, Tae-Wook Chun.

**Project administration:** Gwynne Roth.

**Resources:** Susan Moir.

**Supervision:** Peter D. Sun.

**Writing – original draft:** Jason Segura, Peter D. Sun.

**Writing – review & editing:** Jason Segura, Gwynne Roth, Peter D. Sun.

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
