## [Decision Letter · Decision Letter 0]

30 Aug 2022

PONE-D-22-21679HIV-1 release requires Nef-induced caspase activationPLOS ONE

Dear Dr. Sun,

Thank you for submitting your manuscript to PLOS ONE. After careful consideration, we feel that it has merit but does not fully meet PLOS ONE’s publication criteria as it currently stands. Therefore, we invite you to submit a revised version of the manuscript that addresses the points raised during the review process.

Your manuscript has been seen by two reviewers which are both quite critical related to your study. While one reviewer is mainly concerned about data presentation and clarity, the other one questions novelty and feels that the data does not fully support your conclusions. I understand that this manuscript has already undergone revisions at another journal and further contains a lot of densely packaged data. Ultimately, this might have resulted in a loss of clarity and some overinterpretation of results.

You should carefully go over the manuscript and thoroughly revise it. Remove all grad statements, overinterpretations and so on. Carefully stick to the data. You know that POne does not judge research by its potential impact and also publishs confirmatory studies as long as they are technical sound and all conclusions are supported by the data. This should be the red line for you to follow up on.

We look forward to receiving your revised manuscript.

Kind regards,

Michael Schindler

Academic Editor

PLOS ONE

Journal Requirements:

"This work was supported in part by the National Institutes of Health Strategic Fund in HIV/AIDS research from Office of AIDS Research, and by the Intramural Research Program of National Institute of Allergy and Infectious Diseases,

National Institutes of Health."

"All authors were supported in part by the National Institutes of Health Strategic Fund in HIV/AIDS research from Office of AIDS Research to P.S., and by the Intramural Research Program of National Institute of Allergy and Infectious Diseases, National Institutes of Health to P.S under project number AI-000880."

Reviewers' comments:

Reviewer's Responses to Questions

**Comments to the Author**

1. Is the manuscript technically sound, and do the data support the conclusions?

Reviewer #1: Partly

Reviewer #2: Yes

2. Has the statistical analysis been performed appropriately and rigorously? 

Reviewer #1: N/A

Reviewer #2: I Don't Know

3. Have the authors made all data underlying the findings in their manuscript fully available?

Reviewer #1: Yes

Reviewer #2: Yes

4. Is the manuscript presented in an intelligible fashion and written in standard English?

Reviewer #1: Yes

Reviewer #2: Yes

5. Review Comments to the Author

Reviewer #1: Efficient release of progeny virions from infected cells is the last crucial step in the viral replication cycle. Thus, successful pathogens like HIV evolved intricate strategies that allow their particles to efficiently detach from the cellular surface. CD62L, an L-selectin, was previously implicated in initial binding of HIV virions to cell surfaces, conversely the presence of CD62L also prevents efficient shedding of particles.

In this manuscript, Segura and co-workers suggest that the HIV accessory protein Nef induces caspase activation and subsequent CD62L shedding supporting viral release of HIV-1. The experiments indicate that in HIV-1 infected CD4+ T cells caspase activity/apoptosis is enhanced (like previously shown e.g. Rasola et al, JI, 2001; Doitsh et al, Nature, 2014 and many others). Upon using pan-caspase inhibitors QVD-OPH infection rate were reduced. However, key experiments are still required to support the major claims of this manuscript.

There are many mechanistic ‘jumps’ and gaps in the data that result in overinterpretation in the final model. Can the authors show that CD62L is actually cleaved/shed in the process (e.g. by supernatant blots)? How do the authors propose that Caspases are upregulated and then (which is required for many caspases) activated by Nef? It was previously shown that Vpu and Nef downregulate CD62L surface expression independent of Metalloproteases (Vassena et al, JVI, 2015). Is the model proposed by the authors via caspases, dependent or independent of MMPs. This has to be addressed experimentally (e.g. MMP inhibition) and clarified whether similar mechanisms are at play (sequestration of CD62L). Is the benefit of activating caspases(=enhanced release) larger than the drawbacks (=cell death)? Is shedding of CD62L just a side effect of cells undergoing apoptosis, similar to PARP cleavage?

Please confirm results obtained with the pan-caspase inhibitor by performing caspase KO or KD (e.g. by CRISPR-Cas9 KO or siRNA transfections), or does pooling of the individual caspase inhibitors have the same impact as the pan-caspase inhibitor?

I do not see the paradigm shift yet the authors claim in their abstract. Would larger quantities of virus produced upon caspase-kick really outcompete a loss in replication/virus reservoir due to cell death? One major part of the kick and kill approach is to identify and activate dormant virus to selectively target it. How would that be impacted by caspase inhibitors and the proposed model and why should this study challenge that approach – i.e. change the current concept that released virus is easier to target than latent virus?

Minor issues:

- Be more consistent with the cell type used in different assays and motivate the choice of the specific cell type. (Some results are shown for PBMCs only while others are specific for CD4 T cells or CD8, why?)

- Please control for the infectivity of viral stocks (e.g. WT vs Nef- viruses), as Nef defective viruses often are -dependent on the setting- less infectious.

- Fig. 2D-E: perform the FACS stainings in the same cells!

- Replica of experiments to have n=3 donors or independent experiments: e.g. 1F/4E/ (was only one donor in duplicates or triplicates), 6A (two donors)

- Include Viability assay to exclude cytotoxic effect of the different compounds on the primary cells. E.g. for using DMSO and the pan-caspase inhibitor QVD-OPH.

- Please optimize the figure legends: Including steps of the method which are important to understand the result.

- Please improve the labeling of the figures; e.g. for the Western Blot in Fig. 4F the sizes of the bands are missing and for Fig. 5 the scale bar is missing as well as labelling above about the origin of the picture (TEM/EM)

- Fig 5D: Can the authors identify the structures labeled with yellow arrows beyond claiming they could be virus-like particles? How would HIV stuck to the surface by shedding resistant CD62L look like?

- Overview figure needs to be improved (!): labelling should be Casp and not Cas, the mitochondria is not identifiable. Why does it show HIV-1 entry by endocytosis in a double-layered vesicles?

- Please show for the FACS data (Fig. 2A) single stainings

Reviewer #2: The manuscript proposes a novel and interesting role of caspases in the life cycle of HIV-1, especially important in primary CD4+ cells. I do not have major objections about the experiments and I find the results intriguing. Having read the manuscript, the authors convinced me that the requirement of caspases activity for the release and infectivity of HIV is certainly a concept that deserves more attention and further experiments that this manuscript will certainly stimulate.

However, this is a very dense and massive manuscript which requires an effort to follow. My only request is that authors would make it easier for the readers to follow and understand the experiments, especially by paying attention to the way they formulate the figure legends. Most figure legends are written as if they were a summary of the results, rather than taking care of providing the technical information necessary to understand what kind of data the various graphs represent. Here are some suggestions, but I encourage the authors to revise all legends accordingly.

Figure 1A. Please state the meaning of UI in the legend. Also it is not clear what “Results are pooled from duplicates of conditions from two independent experiments” means in terms of how statistic was assessed. How was t-test calculated from technical or biological repeats? Please explain.

Figure 3. The legend is a succinct result section rather than an explanation of what the data shown in the figure are and how they were acquired. I need to be able to understand what experimental technique was used. I assume that these are data mostly from flow cytometry. It is important to say it, to briefly explain what the data are, which technique was used to acquire them.

Figure 4:

For fig 4A see same comment as figure 3

Fig 4C: can you explain the normalization? Are these copies of viral RNA normalized per ug of gDNA or what else?

Fig 4E. Relative gene expression: relative to what? The reader needs to know.

Figure 5H: “Data taken from triplicate conditions per dose for each treatment and is representative of two independent experiments”. This is really difficult to understand. So, this is a representative experiment and the statistics are on technical replicates?

Figure 6A: please state if this is from RNAseq or RT-PCR

Figure 7. X axis: "days" from what?

Y axis:. Virion associated HIV RNA. How have the authors ensured that this RNA is virion-associated? Were virions pelleted on sucrose to enrich for viral particles? Would it be more appropriate to label the axis “released HIV RNA”?

6. PLOS authors have the option to publish the peer review history of their article (what does this mean?). If published, this will include your full peer review and any attached files.

Reviewer #1: No

Reviewer #2: No

---

## [Author Response · Author response to Decision Letter 0]

14 Dec 2022

We also included raw images for the western blot experiments (Figure 5I). The RNAseq data presented in the manuscript have been deposited to NCBI’s GEO database and are being processed. The rest of the data are presented in the supplemental information.

---

## [Decision Letter · Decision Letter 1]

17 Jan 2023

HIV-1 release requires Nef-induced caspase activation

PONE-D-22-21679R1

Dear Dr. Sun,

We’re pleased to inform you that your manuscript has been judged scientifically suitable for publication and will be formally accepted for publication once it meets all outstanding technical requirements.

Kind regards,

Michael Schindler

Academic Editor

PLOS ONE

Additional Editor Comments (optional):

Reviewers' comments:

Reviewer's Responses to Questions

**Comments to the Author**

1. If the authors have adequately addressed your comments raised in a previous round of review and you feel that this manuscript is now acceptable for publication, you may indicate that here to bypass the “Comments to the Author” section, enter your conflict of interest statement in the “Confidential to Editor” section, and submit your "Accept" recommendation.

Reviewer #2: All comments have been addressed

2. Is the manuscript technically sound, and do the data support the conclusions?

Reviewer #2: Yes

3. Has the statistical analysis been performed appropriately and rigorously? 

Reviewer #2: Yes

4. Have the authors made all data underlying the findings in their manuscript fully available?

Reviewer #2: Yes

5. Is the manuscript presented in an intelligible fashion and written in standard English?

Reviewer #2: Yes

6. Review Comments to the Author

Reviewer #2: (No Response)

7. PLOS authors have the option to publish the peer review history of their article (what does this mean?). If published, this will include your full peer review and any attached files.

Reviewer #2: No

---

## [Editor Report · Acceptance letter]

2 Feb 2023

PONE-D-22-21679R1 

HIV-1 Release Requires Nef-induced Caspase Activation 

Dear Dr. Sun:

I'm pleased to inform you that your manuscript has been deemed suitable for publication in PLOS ONE. Congratulations! Your manuscript is now with our production department. 

Kind regards, 

on behalf of

Prof. Dr. Michael Schindler 

Academic Editor

PLOS ONE